# OD-LoRA: Overcoming the Dilemma between Weight Representation and Gradient Approximation in Low-Rank Adaptation

## Abstract

Low-Rank Adaptation (LoRA) enables efficient adaptation of large pre-trained models to downstream tasks by representing weight updates with trainable low-rank (LR) matrices. Recent studies have shown a different perspective that learning with LoRA is equivalent to using a low-rank approximation of the full fine-tuning gradient, obtained by mapping it onto low-rank subspaces through the LR matrices. In this paper, we theoretically show that LoRA faces a dilemma between these two perspectives: weight update representation and gradient approximation. We first demonstrate that the quality of gradient approximation is improved if the LR matrices have uniform singular values, since non-uniform singular values anisotropically distort the projection of the full gradient onto the subspaces. However, this condition entails a strict constraint on the weight updates, significantly compromising their representational capacity. To **O**vercome this **D**ilemma, we introduce a new method, named OD-LoRA, which decouples the approximated gradient from the singular values of the LR matrices. Specifically, OD-LoRA ensures that the full gradient is mapped through the orthonormal bases of the low-rank subspaces defined by LR matrices, achieving perfect projection onto the subspaces, while still allowing the singular values to represent the weight updates. Consequently, OD-LoRA achieves both the optimal condition for accurate gradient approximation and unconstrained representation of weight updates simultaneously. The experimental results on natural language and vision benchmarks demonstrate that OD-LoRA improves loss convergence and gradient approximation quality, significantly enhancing the adaptation performance of LoRA.

## 1 Introduction

Large-scale pretrained Transformer (Vaswani et al., 2017) models have achieved remarkable success across a wide range of domains, including natural language processing and computer vision. With massive datasets and extensive computational resources, these models are trained to capture rich representations that can be effectively fine-tuned and transferred to diverse downstream tasks. For instance, models such as GPT (Radford et al., 2018), LLaMA (Touvron et al., 2023), BERT (Devlin et al., 2019), and Vision Transformers (ViTs) (Dosovitskiy et al., 2021) have set new benchmarks in language understanding, text generation, and image classification. This pretraining-and-adaptation paradigm has become the standard in the field of artificial intelligence, reducing the need for training models from scratch and driving rapid progress in both research and real-world applications.

However, due to the massive number of parameters, adapting those pretrained models through fine-tuning is often computationally expensive and memory-intensive. For example, GPT-3 (Brown et al., 2020) has about 175 billion parameters, and the parameters of LLaMA-3 models (Grattafiori et al., 2024) range from 8 to 405 billion, making their full fine-tuning impractical. Upon the observation that large pretrained models only require low-rank updates for adaptation (Gooneratne et al., 2020), Low-Rank Adaptation (LoRA) (Hu et al., 2022) addresses the challenge by representing weight updates for adaptation with trainable low-rank matrices (LoRA matrices), reducing the number of trainable parameters to about 1% of the total. Remarkably, LoRA not only demonstrates effective adaptation performance in various natural language and vision benchmarks but also enables the easy merging or detaching of LoRA matrices from the pretrained model, allowing for modular usage.

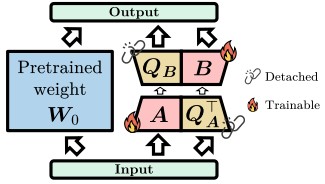

(a) Impact of singular values $\sigma_i$ on low-rank gradient approximation

(b) Illustration of OD-LoRA

Figure 1: **(a)** Learning dynamics in LoRA are equivalent to learning with a low-rank approximation ($\tilde{G}$) of a full fine-tuning gradient ($G$) that is obtained by mapping $G$ onto low-rank subspaces through the LoRA matrices ($A$ and $B$). Non-uniform singular values of $A$ and $B$ distort the projection of $G$ ($G^*$) by anisotropically scaling it. By contrast, when the singular values are uniform, the direction of $G^*$ is preserved, resulting in $\tilde{G}$ with the smallest angular distance from $G$ ($\alpha > \beta$). **(b)** OD-LoRA leverages the orthonormal bases ($Q_A$ and $Q_B$) of the subspaces defined by $A$ and $B$. The computation process of these bases is not accounted for in the gradient computation. By multiplying $A$ with $Q_B$ and $B$ with $Q_A^\top$, the gradient approximation is decoupled from the singular values of $A$ and $B$, while they still represent the weight updates. This allows OD-LoRA to achieve unconstrained weight update representation and accurate gradient approximation simultaneously.

Due to its advantages, LoRA has inspired numerous follow-up studies. Among them, several recent works (Wang et al., 2024; Zhang et al., 2025; Wang et al., 2025) view LoRA from the perspective of low-rank gradient approximation: the learning dynamics in LoRA are equivalent to full fine-tuning with a low-rank approximation of the full gradient. Specifically, the approximated gradient is obtained by mapping the full gradient onto the low-rank subspaces defined by the LoRA matrices. These works propose initialization strategies for the LoRA matrices that leverage the first-step full gradient or adjust the approximated gradient to reduce the approximation error. However, they either overlook the gradient approximation error after the first iteration or incur substantial overhead during training.

In this paper, we provide new insights into LoRA from the perspective of low-rank gradient approximation. Specifically, we prove that, given the subspaces defined by the LoRA matrices, the gradient approximation quality is optimized if the singular values of the LoRA matrices are uniform. As illustrated in Figure 1a, the non-uniform singular values result in anisotropic scaling on the projection of the full gradient onto the low-rank subspaces defined by the LoRA matrices, distorting the projection. By contrast, when the singular values are uniform, the low-rank gradient approximation reduces to a scaled projection, achieving the maximum alignment with the full gradient. However, we show that the uniform singular values of the LoRA matrices compel the non-zero singular values of weight updates to also be uniform, which significantly limits their representational capacity. Therefore, we argue that LoRA faces a dilemma between the representational capacity of weight updates and achieving accurate gradient approximation.

We attribute this dilemma to the dependence of the gradient approximation on the singular values of the LoRA matrices. Thus, to **O**vercome the **D**ilemma in LoRA, we propose a new formulation of low-rank weight updates, called OD-LoRA, which eliminates this dependence by leveraging the orthonormal bases of the subspaces defined by the LoRA matrices. As illustrated in Figure 1b, OD-LoRA treats these bases as constant in gradient computation and multiplies each LoRA matrix with the bases of the other LoRA matrix. We theoretically show that this formulation makes the low-rank gradient approximation independent of the singular values of the LoRA matrices, preserving the direction of the full gradient projection. Simultaneously, the singular values remain responsible for representing the weight updates, thereby retaining their full representational capacity. Furthermore, because these bases are often unstable due to the small norm of the LoRA matrices early in training, we propose initializing them through a few training iterations. Through experiments on various benchmarks and pretrained models, we demonstrate that OD-LoRA converges faster to a lower loss and outperforms LoRA and its variants with only minor overhead in terms of memory and time. We summarize the contribution of our work as follows:

- We show that the gradient approximation quality in LoRA is improved if the LoRA matrices have uniform singular values, while leading to a strict constraint on the weight updates.
- To overcome this dilemma, we propose a new method, OD-LoRA, that achieves both accurate gradient approximation and unconstrained weight update representation simultaneously.
- OD-LoRA improves loss convergence and the alignment between the full gradient and its approximation, resulting in better performance than LoRA and existing methods.

## 2 RELATED WORKS

**Low-Rank Adaptation (LoRA).** To reduce the computational cost in fine-tuning large pretrained models, several parameter-efficient methods have been proposed, including adapter-based methods (Houlsby et al., 2019; He et al., 2022), prefix- or prompt-tuning (Li & Liang, 2021; Lester et al., 2021), and selective update by searching for salient parameters Guo et al. (2021); Sung et al. (2021). While effective, these methods either introduce additional parameters at inference or rely heavily on carefully designed search criteria. On the other hand, LoRA (Hu et al., 2022) introduces trainable low-rank matrices to approximate weight updates, which can be seamlessly merged with the pre-trained weights at inference. Its advantages and strong performance have motivated numerous follow-up studies Kalajdzievski (2023); Hayou et al. (2024b); Si et al. (2025). Several works propose alternative formulations of low-rank weight updates to further improve the performance–efficiency trade-off (Liu et al., 2024; Kopiczko et al., 2024; Lingam et al., 2024; Albert et al., 2025; Koohpayegani et al., 2024). Other research directions include developing effective initialization strategies for the trainable low-rank matrices (Hayou et al., 2024a; Wang et al., 2024; Meng et al., 2024; Zhang et al., 2025; Li et al., 2025), improving the rank of weight updates (Huang et al., 2025; Lialin et al., 2024), and designing layer-wise adaptive rank search methods (Zhang et al., 2023; Ke et al., 2024).

**Low-Rank Gradient Approximation.** There have been attempts to reduce the computational cost during training by approximating the original gradient in a low-rank subspace (Gooneratne et al., 2020; Zhao et al., 2024). Several recent studies have examined LoRA from the perspective of low-rank gradient approximation (Wang et al., 2024; Zhang et al., 2025; Wang et al., 2025; Zhang & Pilanci, 2024; Yu et al., 2025). Wang et al. (2024); Zhang et al. (2025) propose initialization strategies for LoRA that minimize the gradient approximation error at the first training step. Similar to our approach, Wang et al. (2025); Zhang & Pilanci (2024); Yu et al. (2025) aim to optimize the update geometry via preconditioning. However, as discussed in Section D, such methods tend to amplify useless gradients when the singular value distribution of LoRA matrices is skewed, affecting the training dynamics. By contrast, our method structurally decouples the gradient approximation from the spectral properties of LoRA matrices, ensuring more robust training dynamics.

## 3 PROPOSED METHOD

### 3.1 PRELIMINARIES

**Low-Rank Weight Update Representation.** Let $\boldsymbol{W}_0 \in \mathbb{R}^{d_{\text{out}} \times d_{\text{in}}}$ denote the weight matrix of a linear layer in a pretrained model, with input dimension $d_{\text{in}}$ and output dimension $d_{\text{out}}$. To adapt the model to a downstream task, we train it to learn task-specific weight updates. Instead of updating all the parameters of $\boldsymbol{W}_0$, Low-Rank Adaptation (LoRA) assumes the low-rank structure of desired weight updates Gooneratne et al. (2020) and learns a low-rank decomposition of the updates:

$$\boldsymbol{W}_{\text{eff}} = \boldsymbol{W}_0 + \Delta\boldsymbol{W}_{\text{LoRA}} = \boldsymbol{W}_0 + s\boldsymbol{B}\boldsymbol{A}, \tag{1}$$

where $\boldsymbol{A} = [\boldsymbol{a}_1, \boldsymbol{a}_2, \ldots, \boldsymbol{a}_r]^\top \in \mathbb{R}^{r \times d_{\text{in}}}$ and $\boldsymbol{B} = [\boldsymbol{b}_1, \boldsymbol{b}_2, \ldots, \boldsymbol{b}_r] \in \mathbb{R}^{d_{\text{out}} \times r}$ are trainable low-rank matrices and $s$ is a scaling factor. For one layer, the total number of trainable parameters in LoRA is $r(d_{\text{out}} + d_{\text{in}})$, which is much smaller than that of full-parameter training case, $d_{\text{out}} \times d_{\text{in}}$, if $r \ll \min(d_{\text{out}}, d_{\text{in}})$. It is common to initialize $\boldsymbol{A}$ from a random Gaussian or uniform distribution and to initialize $\boldsymbol{B}$ to zero, ensuring that the model initially behaves identically to the pretrained one. Once $\boldsymbol{A}$ and $\boldsymbol{B}$ are trained, they can be easily merged with or detached from the pretrained weights.

**Low-Rank Gradient Approximation Perspective.** Similar to Wang et al. (2024); Zhang et al. (2025); Wang et al. (2025), we view LoRA from the low-rank gradient approximation perspective. Let $\boldsymbol{G} = \frac{\partial \mathcal{L}}{\partial \boldsymbol{W}_{\text{eff}}}$ denote the gradient of loss $\mathcal{L}$ with respect to $\boldsymbol{W}_{\text{eff}}$. We refer to $\boldsymbol{G}$ as *full gradient* since it is used in a full-parameter fine-tuning scenario. The gradient with respect to $\boldsymbol{A}$ and $\boldsymbol{B}$ is computed as $\frac{\partial \mathcal{L}}{\partial \boldsymbol{A}} = s\boldsymbol{B}^\top \boldsymbol{G}$ and $\frac{\partial \mathcal{L}}{\partial \boldsymbol{B}} = s\boldsymbol{G}\boldsymbol{A}^\top$. Then, the differential of $\boldsymbol{A}$ and $\boldsymbol{B}$ in the context of gradient descent with a learning rate $\eta$ are given by $d\boldsymbol{A} = -\eta\frac{\partial \mathcal{L}}{\partial \boldsymbol{A}} = -\eta s\boldsymbol{B}^\top \boldsymbol{G}$ and $d\boldsymbol{B} = -\eta\frac{\partial \mathcal{L}}{\partial \boldsymbol{B}} = -\eta s\boldsymbol{G}\boldsymbol{A}^\top$. Consequently, the differential of $\boldsymbol{W}_{\text{eff}}$ is expressed as

$$d\boldsymbol{W}_{\text{eff}} = s(d\boldsymbol{B}\,\boldsymbol{A} + \boldsymbol{B}\,d\boldsymbol{A}) = -\eta s^2(\boldsymbol{G}\boldsymbol{A}^\top\boldsymbol{A} + \boldsymbol{B}\boldsymbol{B}^\top\boldsymbol{G}) = -\eta\tilde{\boldsymbol{G}}. \tag{2}$$

This implies that LoRA is equivalent to full fine-tuning $\boldsymbol{W}_{\text{eff}}$ using the equivalent gradient $\tilde{\boldsymbol{G}}$. Let $\mathcal{R}(\boldsymbol{A})$ and $\mathcal{C}(\boldsymbol{B})$ denote the row space of $\boldsymbol{A}$ and the column space of $\boldsymbol{B}$, respectively. Then, $\boldsymbol{G}\boldsymbol{A}^\top\boldsymbol{A}$ and $\boldsymbol{B}\boldsymbol{B}^\top\boldsymbol{G}$ can be interpreted as the mapping of the rows of $\boldsymbol{G}$ onto $\mathcal{R}(\boldsymbol{A})$ and the mapping of columns of $\boldsymbol{G}$ onto $\mathcal{C}(\boldsymbol{B})$. Since these subspaces are low rank, $\tilde{\boldsymbol{G}}$ can be interpreted as a low-rank approximation of $\boldsymbol{G}$.

## 3.2 Dilemma in LoRA: Weight Representation vs. Gradient Approximation

Let $\mathcal{W} = \{\boldsymbol{W}'^l = \boldsymbol{W}_0^l + s\boldsymbol{B}^l\boldsymbol{A}^l\}_{l=1}^L$ denote the set of the effective weights across $L$ layers. Let $\mathcal{G} = \{\boldsymbol{G}^l\}_{l=1}^L$ and $\tilde{\mathcal{G}} = \{\tilde{\boldsymbol{G}}^l\}_{l=1}^L$ denote the full gradient of loss $\mathcal{L}$ with respect to $\mathcal{W}$ and its low-rank approximation, respectively. Assuming that $\mathcal{L}$ is $\beta$-smooth, we can derive the following inequality on the loss decrease guarantee when $\mathcal{W}$ is updated using $\tilde{\mathcal{G}}$:

$$\mathcal{L}(\mathcal{W} - \eta\tilde{\mathcal{G}}) - \mathcal{L}(\mathcal{W}) \leq -\eta\langle\mathcal{G}, \tilde{\mathcal{G}}\rangle + \frac{\beta}{2}\eta^2\|\tilde{\mathcal{G}}\|^2. \tag{3}$$

Here, $\langle\mathcal{G}, \tilde{\mathcal{G}}\rangle \coloneqq \sum_{l=1}^L \langle\boldsymbol{G}^l, \tilde{\boldsymbol{G}}^l\rangle_F$ and $\|\mathcal{G}\| = \sqrt{\langle\mathcal{G}, \mathcal{G}\rangle}$ where $\langle\cdot, \cdot\rangle_F$ denotes the Frobenius inner product. Details are provided in Appendix E. Equation 3 suggests that for a sufficiently small $\eta$, the approximated gradient will result in better loss convergence as $\langle\mathcal{G}, \tilde{\mathcal{G}}\rangle$ increases. Thus, in contrast to prior works that define gradient approximation error as $\|\mathcal{G} - \tilde{\mathcal{G}}\|$ (Wang et al., 2024; Zhang et al., 2025; Wang et al., 2025), we propose using $\langle\mathcal{G}, \tilde{\mathcal{G}}\rangle$ as a measure of the approximation quality of $\tilde{\mathcal{G}}$. We derive the following theorem on the conditions required for the LoRA matrices to improve the approximation quality:

---

**Theorem 1: Optimal Condition for Improving Gradient Approximation Quality in LoRA**

*Consider the gradient approximation quality in LoRA:*

$$\langle\mathcal{G}, \tilde{\mathcal{G}}\rangle = \sum_{l=1}^L \langle\boldsymbol{G}^l, \tilde{\boldsymbol{G}}^l\rangle_F = s^2 \sum_{l=1}^L \langle\boldsymbol{G}^l, \boldsymbol{G}^l\boldsymbol{A}^{l\top}\boldsymbol{A}^l + \boldsymbol{B}^l\boldsymbol{B}^{l\top}\boldsymbol{G}^l\rangle_F.$$

*Assume the energy of $\boldsymbol{A}^l$ and $\boldsymbol{B}^l$, the row space of $\boldsymbol{A}^l$, and the column space of $\boldsymbol{B}^l$ are given for all $l$. Then, $\langle\mathcal{G}, \tilde{\mathcal{G}}\rangle$ is maximized if $\boldsymbol{A}^l$ and $\boldsymbol{B}^l$ have uniform singular values for all $l$.*

---

See Section A for the proof. Let $\boldsymbol{A} = \boldsymbol{U}_A\boldsymbol{\Sigma}_A\boldsymbol{V}_A^\top$ and $\boldsymbol{B} = \boldsymbol{U}_B\boldsymbol{\Sigma}_B\boldsymbol{V}_B^\top$ be the compact singular value decomposition (SVD) of $\boldsymbol{A}$ and $\boldsymbol{B}$. To examine how the distribution of singular values of $\boldsymbol{A}$ and $\boldsymbol{B}$ affects the gradient approximation quality, we rewrite $\tilde{\mathcal{G}}$ as $\boldsymbol{G}\boldsymbol{A}^\top\boldsymbol{A} + \boldsymbol{B}\boldsymbol{B}^\top\boldsymbol{G} = \boldsymbol{G}\boldsymbol{V}_A\boldsymbol{\Sigma}_A^2\boldsymbol{V}_A^\top + \boldsymbol{U}_B\boldsymbol{\Sigma}_B^2\boldsymbol{U}_B^\top\boldsymbol{G}$. Without the singular value terms, $\boldsymbol{G}\boldsymbol{V}_A\boldsymbol{V}_A^\top$ and $\boldsymbol{U}_B\boldsymbol{U}_B^\top\boldsymbol{G}$ are exactly the orthogonal projections of the rows of $\boldsymbol{G}$ onto $\mathcal{R}(\boldsymbol{A})$ and the columns of $\boldsymbol{G}$ onto $\mathcal{C}(\boldsymbol{B})$, respectively. This implies that the singular values of $\boldsymbol{A}$ and $\boldsymbol{B}$ act as weights on the components of the projection, which are aligned with the corresponding singular vectors. As illustrated in Figure 1a, if the singular values are non-uniform, they anisotropically scale the projected components, distorting the projection of $\boldsymbol{G}$ onto the subspaces. On the other hand, if $\boldsymbol{A}$ and $\boldsymbol{B}$ have uniform singular values, the projection is scaled uniformly in all directions, yielding $\tilde{\mathcal{G}}$ that is maximally aligned with $\mathcal{G}$. The proof shows that given the energy assumption on $\boldsymbol{A}$ and $\boldsymbol{B}$, this maximal alignment result in maximizing $\langle\boldsymbol{G}, \tilde{\boldsymbol{G}}\rangle_F$.

However, in the context of the LoRA formulation $\Delta\boldsymbol{W}_{\text{LoRA}} = s\boldsymbol{B}\boldsymbol{A}$, the uniform singular value condition may impose an overly restrictive constraint on the representational capacity. Motivated by the universal approximation theorem (Hornik et al., 1989), we formally define the representational capacity of rank-$r$ weight updates $\Delta\boldsymbol{W}$ as its capability to represent an arbitrary rank-$r$ matrix:

**Definition 1** (Representational capacity). For a rank-$r$ matrix $\boldsymbol{X} \in \mathbb{R}^{d_{\text{out}} \times d_{\text{in}}}$, we define the representational capacity of $\Delta\boldsymbol{W}$ as the reciprocal of the minimum achievable error between $\boldsymbol{X}$ and $\Delta\boldsymbol{W}$:

$$\text{Cap}_{\boldsymbol{X}}(\Delta\boldsymbol{W}) \coloneqq \frac{1}{\min_{\Delta\boldsymbol{W}}\|\boldsymbol{X} - \Delta\boldsymbol{W}\|_F^2}.$$

While $\text{Cap}_{\boldsymbol{X}}(\Delta\boldsymbol{W})$ can be further quantified through an optimization or expectation over $\boldsymbol{X}$, this is unnecessary in this paper. Using this definition, we prove that LoRA faces a dilemma between the representational capacity $\text{Cap}_{\boldsymbol{X}}(\Delta\boldsymbol{W}_{\text{LoRA}})$ and the gradient approximation quality:

> **Theorem 2: Dilemma in LoRA: Representational Capacity or Gradient Approximation Quality?**
>
> *The formulation of LoRA, $\Delta W_{LoRA} = sBA$ where both $A$ and $B$ are rank-$r$, faces the following dilemma between representational capacity and gradient approximation quality:*
> - ***Full representational capacity but suboptimal gradient approximation.*** *Without any constraints, $Cap_X(\Delta W_{LoRA}) = \infty$, but this requires either $A$ or $B$ to have non-uniform singular values, which impedes improving the gradient approximation quality in Theorem 1.*
> - ***Accurate gradient approximation but limited representational capacity.*** *If both $A$ and $B$ have uniform singular values (satisfying the optimal condition of Theorem 1), the non-zero singular values of $\Delta W_{LoRA}$ are uniform, significantly limiting the representational capacity: $Cap_X(\Delta W_{LoRA}) = 1/\sum_{i=1}^{r}(\sigma_i - \frac{1}{r}\sum_{i=1}^{r}\sigma_i)^2$, where $X \in \mathbb{R}^{d_{out} \times d_{in}}$ is an arbitrary rank-$r$ matrix with non-zero singular values $\{\sigma_1, \sigma_2, \ldots, \sigma_r\}$,*

See Section B for the proof. Theorem 2 shows that the LoRA formulation is unable to achieve both optimal gradient approximation quality and full representational capacity at the same time. Notably, under the uniform singular value condition, the representational capacity is reduced to $1/\sum_{i=1}^{r}(\sigma_i - \frac{1}{r}\sum_{i=1}^{r}\sigma_i)^2$ where the denominator is proportional to the variance of the singular values of a target rank-$r$ matrix. This suggests that if the desired weight updates favor singular value spectrum with a high variance, the achievable minimum error between $\Delta W_{\text{LoRA}}$ and the desired updates is large under the uniform singular value condition. Indeed, Figure 5 shows that the weight updates obtained via full fine-tuning typically exhibit highly skewed singular values, suggesting that the uniform singular value condition may hinder the representation of desired weight updates.

## 3.3 OD-LoRA: Decoupling the Gradient Approximation from Singular Values

We argue that this dilemma arises because in the formulation of LoRA, both the weight updates and the approximated gradient depend on the singular values of $A$ and $B$. Thus, to **O**vercome the **D**ilemma, we propose a new LoRA formulation, named OD-LoRA, which decouples the low-rank approximation of the full gradient from the singular values of $A$ and $B$, while preserving the maximal representational capacity of the weight updates. To do so, we leverage the orthonormal bases of $\mathcal{R}(A)$ and $\mathcal{C}(B)$ since they are independent of the singular values of $A$ and $B$. Specifically, we compute the orthonormal bases via QR decomposition: $A^{\top} = Q_A R_A$ and $B = Q_B R_B$ where the columns of $Q_A$ and $Q_B$ are the orthonormal bases of $\mathcal{R}(A)$ and $\mathcal{C}(B)$. We then formulate the weight updates so that the gradients for $A$ and $B$ depend only on these bases, while the singular values of $A$ and $B$ remain responsible for representing the weight updates:

$$\Delta W_{\text{OD-LoRA}} = s(Q_B A + B Q_A^{\top}) = s[Q_B \ B]\begin{bmatrix} A \\ Q_A^{\top} \end{bmatrix}, \tag{4}$$

During the backward pass, we treat $Q_A$ and $Q_B$ as constants. Hence, the gradient with respect to $A$ and $B$ is expressed in terms of $Q_B$ and $Q_A$, respectively: $\frac{\partial \mathcal{L}}{\partial A} = s Q_B^{\top} G$ and $\frac{\partial \mathcal{L}}{\partial B} = s G Q_A$. Thus, the singular values of $A$ and $B$ need not be constrained to satisfy the optimal condition of Theorem 1, allowing unconstrained representation of $\Delta W_{\text{OD-LoRA}}$. We theoretically show that our new formulation address the dilemma in LoRA:

> **Theorem 3: OD-LoRA Overcomes the Dilemma in LoRA**
>
> *The formulation of OD-LoRA in Equation 4 possesses the following properties simultaneously.*
>
> *1. Define $W_{eff} = W_0 + \Delta W_{OD\text{-}LoRA}$. Then, the low-rank approximation of full gradient $G$ is given by*
> $$\tilde{G} = s^2(G Q_A Q_A^{\top} + Q_B Q_B^{\top} G),$$
> *where $Q_A$ and $Q_B$ have uniform singular values (all equal to 1), satisfying the optimal condition for improving the gradient approximation quality in Theorem 1.*
>
> *2. For an arbitrary rank-$r$ matrix $X \in \mathbb{R}^{d_{out} \times d_{in}}$, $Cap_X(\Delta W_{OD\text{-}LoRA}) = \infty$, implying $\Delta W_{OD\text{-}LoRA}$ can represent any rank-$r$ matrices.*

See Section C for the proof. Theorem 3 demonstrates that OD-LoRA is able to achieve both the accurate low-rank approximation of the full gradient and the maximum representational capacity of weight updates simultaneously, overcoming the dilemma in LoRA.

---

**Algorithm 1:** Training Process of OD-LoRA

---

**Input:** pretrained weight $\boldsymbol{W}_0$, training dataset $\mathcal{D} = \{\mathcal{B}_1, \ldots \mathcal{B}_{T_{\text{total}}}\}$, and $\boldsymbol{A}$ and $\boldsymbol{B}$ s.t. $\boldsymbol{B}\boldsymbol{A} = \boldsymbol{0}$

// Initialize $\boldsymbol{Q_A}$, and $\boldsymbol{Q_B}$

**for** $t = 1$ **to** $N \times T$ **do**
    **if** $t \leq T$ **then**
        $\llcorner$ $\Delta\boldsymbol{W} \leftarrow s\boldsymbol{B}\boldsymbol{A}$
    **else**
        $\Delta\boldsymbol{W} \leftarrow s[\boldsymbol{Q_B}\ \boldsymbol{B}] \begin{bmatrix} \boldsymbol{A} \\ \boldsymbol{Q_A^\top} \end{bmatrix}$ // Equation 4
    Forward with $\mathcal{B}_t$ using $\boldsymbol{W}_{\text{eff}} = \boldsymbol{W}_0 + \Delta\boldsymbol{W}$
    Backward and update $\boldsymbol{A}$ and $\boldsymbol{B}$
    **if** $t \bmod T = 0$ **then**
        Compute the rank-$r$ truncated SVD of $\Delta\boldsymbol{W}$: $\Delta\boldsymbol{W} = \boldsymbol{U}_r\boldsymbol{\Sigma}_r\boldsymbol{V}_r^\top$
        $\boldsymbol{A} \leftarrow \boldsymbol{0},\ \boldsymbol{B} \leftarrow \boldsymbol{0},\ \boldsymbol{Q_A} \leftarrow \boldsymbol{V}_r,\ \boldsymbol{Q_B} \leftarrow \boldsymbol{U}_r$
        Clear all optimizer states

// Currently, $\boldsymbol{A} = \boldsymbol{0}$, $\boldsymbol{B} = \boldsymbol{0}$, $\boldsymbol{Q_A} = \boldsymbol{V}_r$, $\boldsymbol{Q_B} = \boldsymbol{U}_r$

**for** $t = 1$ **to** $T_{\text{total}}$ **do**
    $\Delta\boldsymbol{W} \leftarrow s[\boldsymbol{Q_B}\ \boldsymbol{B}] \begin{bmatrix} \boldsymbol{A} \\ \boldsymbol{Q_A^\top} \end{bmatrix}$
    Forward with $\mathcal{B}_t$ using $\boldsymbol{W}_{\text{eff}} = \boldsymbol{W}_0 + \Delta\boldsymbol{W}$
    Backward and update $\boldsymbol{A}$ and $\boldsymbol{B}$
    **if** $t = T$ **then**
        // Ensure $\boldsymbol{Q_A}$ and $\boldsymbol{Q_B}$ form bases for $\mathcal{R}(\boldsymbol{A})$ and $\mathcal{C}(B)$
        Compute the rank-$r$ truncated SVD of $\Delta\boldsymbol{W}$: $\Delta\boldsymbol{W} = \boldsymbol{U}_r\boldsymbol{\Sigma}_r\boldsymbol{V}_r^\top$
        $\boldsymbol{A} \leftarrow \frac{1}{2s}\boldsymbol{\Sigma}_r\boldsymbol{V}_r^\top,\ \boldsymbol{B} \leftarrow \frac{1}{2s}\boldsymbol{U}_r\boldsymbol{\Sigma}_r,\ \boldsymbol{Q_A} \leftarrow \boldsymbol{V}_r,\ \boldsymbol{Q_B} \leftarrow \boldsymbol{U}_r$ // Equation 5
        Clear all optimizer states
    **else if** $t > T$ and $t \bmod \left\lceil \frac{5t}{T_{\text{total}}} \right\rceil$ **then**
        // Adjust the update interval according to $t$
        Compute the QR decomposition of $\boldsymbol{A}^\top$ and $\boldsymbol{B}$: $\boldsymbol{A}^\top = \boldsymbol{Q'_A}\boldsymbol{R'_A}$ and $\boldsymbol{B} = \boldsymbol{Q'_B}\boldsymbol{R'_B}$
        $\boldsymbol{Q_A} \leftarrow \boldsymbol{Q'_A},\ \leftarrow \boldsymbol{Q_B} \leftarrow \boldsymbol{Q'_B}$

**return** $\boldsymbol{A}, \boldsymbol{B}$

---

At the beginning of training, $\Delta\boldsymbol{W}_{\text{OD-LoRA}}$ must be zero to ensure that learning starts from the pretrained weight. Thus, during the initial training phase, representing the weight updates as in Equation 4 may be ineffective, since $\mathcal{R}(\boldsymbol{A})$ and $\mathcal{C}(\boldsymbol{B})$ are not yet well-defined and can change rapidly due to small norms. To address this, at the beginning of training, we allow $\boldsymbol{Q_A}$ and $\boldsymbol{Q_B}$ to not coincide with the orthonormal bases of these subspaces and initialize them through a few training iterations. Specifically, we repeat the following processes $N$ times: training $\boldsymbol{A}$ and $\boldsymbol{B}$ for $T$ steps, updating $\boldsymbol{Q_A}$ and $\boldsymbol{Q_B}$ using the top-$r$ right and left singular vectors of the resulting weight updates, and resetting $\boldsymbol{A}$ and $\boldsymbol{B}$ to zero. For the first iteration, we use the LoRA formulation due to the absence of $\boldsymbol{Q_A}$ and $\boldsymbol{Q_B}$, and switch to the OD-LoRA formulation thereafter. Additionally, at the end of each iteration, we clear all optimizer states (i.e., momentum buffers) to eliminate their impact on the next iteration. After these $N \times T$ steps, we obtain $\boldsymbol{Q_A}$ and $\boldsymbol{Q_B}$ that are effectively initialized for the early training stage. However, they are not yet the bases of $\mathcal{R}(\boldsymbol{A})$ and $\mathcal{C}(\boldsymbol{B})$ since $\boldsymbol{A} = \boldsymbol{B} = \boldsymbol{0}$. To address this mismatch, we train $\boldsymbol{A}$ and $\boldsymbol{B}$ using the fixed $\boldsymbol{Q_A}$ and $\boldsymbol{Q_B}$ for the first $T$ iterations, and then set

$$\boldsymbol{A} = \frac{1}{2s}\boldsymbol{\Sigma}_r\boldsymbol{V}_r^\top, \quad \boldsymbol{B} = \frac{1}{2s}\boldsymbol{U}_r\boldsymbol{\Sigma}_r, \quad \boldsymbol{Q_A} = \boldsymbol{V}_r, \quad \boldsymbol{Q_B} = \boldsymbol{U}_r, \tag{5}$$

where $\Delta\boldsymbol{W}_{\text{OD-LoRA}} = \boldsymbol{U}_r\boldsymbol{\Sigma}_r\boldsymbol{V}_r^\top$ is the rank-$r$ truncated SVD. We also reset all optimizer states after this process. By doing so, we ensure that the columns of $\boldsymbol{Q_A}$ and $\boldsymbol{Q_B}$ form the orthonormal bases of $\mathcal{R}(\boldsymbol{A})$ and $\mathcal{C}(\boldsymbol{B})$, respectively. After the initialization phase, we update $\boldsymbol{Q_A}$ and $\boldsymbol{Q_B}$ because $\mathcal{R}(\boldsymbol{A})$ and $\mathcal{C}(\boldsymbol{B})$ change during training. See Algorithm 1 for the overall training process.

**Training overhead.** We set $N$ and $T$ to 5 and 10, respectively, resulting in 6 SVD computations and roughly a 1% increase in training iterations. To reduce the overhead of the QR decomposition, we adjust the update interval for $\boldsymbol{Q_A}$ and $\boldsymbol{Q_B}$. Specifically, we divide training into five phases, setting the update interval to one training step for the first phase to account for the rapid change of the subspaces during early training, and then increasing it by one after each subsequent phase. As a result, the initialization strategy and the updates of $\boldsymbol{Q_A}$ and $\boldsymbol{Q_B}$ incur minor training overhead, as detailed in Section 4.7.

Table 1: **Results on commonsense reasoning tasks.**

| Model | Method | Rank | BoolQ | PIQA | SIQA | HellaSwag | WinoGrande | ARC-c | ARC-e | OBQA | Avg. |
|---|---|---|---|---|---|---|---|---|---|---|---|
| | Full FT. | - | 66.96±0.21 | 80.31±0.12 | 75.85±0.47 | 85.48±0.43 | 75.61±0.67 | 65.16±0.70 | 80.32±0.38 | 76.27±0.34 | 75.75 |
| | LoRA | | 63.94±0.04 | 76.46±0.28 | 70.62±0.36 | 44.50±0.48 | 65.48±0.74 | 57.20±0.28 | 75.06±0.56 | 65.60±1.41 | 64.86 |
| | rsLoRA | | 64.58±0.07 | 77.68±0.56 | 72.65±0.60 | 55.14±1.73 | 69.48±0.15 | 58.70±0.00 | 76.19±0.38 | 69.40±0.57 | 67.98 |
| | LoRA+ | 8 | 63.35±0.00 | 76.82±0.03 | 72.00±0.36 | 77.18±1.01 | 70.90±1.49 | 58.47±0.44 | 76.06±0.02 | 70.20±0.28 | 70.62 |
| | PiSSA | | 64.94±0.20 | 77.49±0.13 | 72.79±0.31 | 59.89±3.06 | 69.27±0.04 | 58.22±0.52 | 76.18±0.12 | 71.33±0.19 | 68.77 |
| Gemma-2B | DoRA | | 64.72±0.16 | 77.57±0.41 | 72.77±0.36 | 56.56±0.61 | 69.82±0.41 | 59.02±0.56 | 75.94±0.08 | 69.73±0.47 | 68.27 |
| | **OD-LoRA** | | **65.85±0.37** | **79.22±0.31** | **74.51±0.43** | **80.34±0.96** | **73.11±0.41** | **62.17±0.20** | **78.77±0.44** | **73.13±0.75** | **73.39** |
| | LoRA | | 64.24±0.10 | 76.22±0.15 | 70.71±0.05 | 45.05±1.40 | 65.98±0.11 | 57.51±0.24 | 74.96±0.18 | 65.13±0.47 | 64.98 |
| | rsLoRA | | 65.63±0.04 | 78.89±0.62 | 73.97±0.17 | 77.69±0.00 | 71.03±0.45 | 61.26±0.72 | 78.55±0.08 | 72.80±1.41 | 72.48 |
| | LoRA+ | 32 | 64.62±0.14 | 78.32±0.10 | 73.18±0.65 | 81.73±0.10 | 71.48±0.45 | 61.59±0.20 | 76.91±0.30 | 73.20±0.28 | 72.63 |
| | PiSSA | | 65.74±0.32 | 77.98±0.05 | 74.14±0.34 | 72.90±1.05 | 70.96±0.33 | 59.47±0.97 | 76.91±0.28 | 74.40±0.28 | 71.56 |
| | DoRA | | 65.30±0.07 | 78.89±0.77 | 74.19±0.68 | 74.17±1.16 | 71.56±0.19 | 61.06±0.68 | 78.56±0.20 | 72.60±1.41 | 72.04 |
| | **OD-LoRA** | | **66.40±0.16** | **80.92±0.28** | **75.04±0.31** | **84.89±0.34** | **74.01±0.15** | **62.97±0.60** | **79.52±0.08** | **75.47±0.38** | **74.90** |
| | Full FT. | - | 75.24±0.21 | 89.70±0.22 | 81.51±0.24 | 96.11±0.15 | 87.66±0.17 | 81.23±0.54 | 92.07±0.19 | 88.02±0.31 | 86.44 |
| | LoRA | | 72.98±0.06 | 87.60±0.31 | 79.67±0.39 | 94.35±0.14 | 83.61±0.15 | 78.95±0.44 | 90.14±0.02 | 83.87±1.04 | 83.89 |
| | rsLoRA | | 73.40±0.19 | 88.23±0.26 | 80.26±0.41 | 95.19±0.43 | 84.98±0.19 | 79.21±0.32 | 90.70±0.12 | 84.80±0.28 | 84.60 |
| | LoRA+ | 8 | 73.07±0.61 | 88.03±0.44 | 80.22±0.29 | 94.75±0.01 | 85.00±0.07 | 78.95±0.36 | 90.16±0.06 | 85.20±0.00 | 84.42 |
| | PiSSA | | 73.68±0.01 | 88.16±0.10 | 80.37±0.19 | 95.20±0.05 | 85.82±0.19 | 80.12±0.60 | 90.36±0.06 | 85.67±0.66 | 84.92 |
| LLaMA3-8B | DoRA | | 73.20±0.01 | 87.85±0.10 | 80.21±0.27 | 95.22±0.00 | 84.37±0.45 | 79.66±0.16 | 90.53±0.12 | 85.53±0.19 | 84.57 |
| | **OD-LoRA** | | **74.02±0.12** | **88.88±0.36** | **81.01±0.22** | **96.06±0.06** | **87.13±0.22** | **79.89±0.56** | **90.88±0.22** | **86.33±0.09** | **85.53** |
| | LoRA | | 73.06±0.13 | 87.45±0.05 | 79.68±0.00 | 94.50±0.08 | 83.32±0.15 | 79.58±0.52 | 90.31±0.02 | 84.07±0.75 | 84.00 |
| | rsLoRA | | 72.56±0.89 | 88.47±0.08 | 80.74±0.39 | 91.80±5.36 | 85.61±0.52 | 79.69±0.60 | 90.90±0.08 | 85.53±1.04 | 84.41 |
| | LoRA+ | 32 | 73.43±0.30 | 88.25±0.10 | 80.03±0.05 | 94.99±0.01 | 85.87±0.41 | 79.92±0.20 | 90.05±0.32 | 85.73±1.04 | 84.78 |
| | PiSSA | | 74.46±0.52 | 89.03±0.44 | 80.79±0.41 | 95.48±0.02 | 86.98±0.11 | 80.12±0.36 | 90.75±0.14 | **86.33±0.66** | 85.49 |
| | DoRA | | 73.86±0.23 | 88.72±0.33 | 80.98±0.27 | 95.67±0.05 | 85.85±0.41 | 79.78±0.48 | 91.08±0.00 | 85.87±0.09 | 85.23 |
| | **OD-LoRA** | | **74.65±0.09** | **89.48±0.05** | **81.01±0.07** | **96.11±0.00** | **88.08±0.56** | **81.06±0.36** | **92.02±0.22** | 85.93±1.04 | **86.04** |

Table 2: **Results on natural language generation tasks.**

| Method | Rank | Gemma-2B | | | LLaMA3-8B | | |
|---|---|---|---|---|---|---|---|
| | | MATH | GSM8K | HumanEval | MATH | GSM8K | HumanEval |
| Full FT. | - | 19.17±0.22 | 56.23±0.23 | 33.35±0.65 | 26.47±0.41 | 77.04±0.18 | 49.11±0.44 |
| LoRA | | 16.12±0.28 | 45.59±0.95 | 27.24±0.57 | 23.68±0.28 | 73.44±0.22 | 42.53±1.80 |
| rsLoRA | | 17.10±0.24 | 49.10±1.44 | 29.27±1.79 | 24.52±0.09 | 75.71±0.23 | 42.23±1.63 |
| LoRA+ | 8 | 17.09±0.46 | 50.30±0.81 | 27.64±0.76 | 24.65±0.63 | 75.27±0.59 | 45.27±2.84 |
| PiSSA | | 16.44±0.38 | 50.97±0.74 | 28.86±0.58 | 24.43±0.23 | 74.96±1.01 | 46.19±2.64 |
| DoRA | | 17.19±0.28 | 50.61±0.85 | 28.35±1.52 | 24.63±0.21 | 75.41±1.32 | 43.29±1.14 |
| **OD-LoRA** | | **17.79±0.18** | **52.08±0.53** | **31.30±0.76** | **25.89±0.30** | **76.62±0.34** | **47.15±1.44** |
| LoRA | | 16.31±0.13 | 45.67±0.91 | 28.66±0.50 | 23.66±0.25 | 73.49±0.59 | 42.53±1.09 |
| rsLoRA | | 18.09±0.04 | 51.93±0.53 | 31.10±1.32 | 25.97±0.17 | 76.28±0.96 | 44.36±1.39 |
| LoRA+ | 32 | 17.45±0.39 | 52.74±0.75 | 31.30±1.25 | 25.01±0.11 | 76.11±0.20 | 46.34±1.29 |
| PiSSA | | 17.23±0.24 | 53.38±0.77 | 32.11±0.76 | 24.80±0.41 | 76.39±0.75 | 47.10±2.38 |
| DoRA | | 18.27±0.14 | 52.41±0.24 | 30.79±1.52 | 25.87±0.42 | 76.42±0.17 | 44.97±1.00 |
| **OD-LoRA** | | **18.47±0.14** | **55.34±1.10** | **34.15±0.50** | **26.45±0.39** | **77.22±0.19** | **47.76±1.25** |

Table 3: **Results on image classification tasks with ViT-Base.** See Section G for details and more results.

| Method | Rank | Cars | CUB200 | SUN397 |
|---|---|---|---|---|
| Full FT. | - | 84.26±0.19 | 86.35±0.18 | 74.76±0.26 |
| LoRA | | 78.66±0.23 | 85.65±0.05 | 74.35±0.28 |
| rsLoRA | | 78.48±0.07 | 85.67±0.41 | 74.63±0.13 |
| LoRA+ | 8 | 79.05±0.25 | 85.28±0.11 | 72.87±0.28 |
| PiSSA | | 78.03±0.32 | 85.17±0.24 | 72.82±0.10 |
| DoRA | | 78.93±0.30 | 85.74±0.35 | 74.46±0.09 |
| **OD-LoRA** | | **80.47±0.14** | **86.08±0.16** | **74.92±0.16** |
| LoRA | | 81.48±0.20 | 85.75±0.28 | 70.82±0.07 |
| rsLoRA | | 80.57±0.22 | 85.42±0.41 | 73.92±0.26 |
| LoRA+ | 32 | 79.53±0.60 | 85.68±0.31 | 67.66±0.06 |
| PiSSA | | 80.43±0.17 | 84.90±0.28 | 69.73±0.12 |
| DoRA | | 81.07±0.03 | 85.67±0.22 | 74.05±0.21 |
| **OD-LoRA** | | **83.04±0.32** | **86.06±0.24** | **74.92±0.21** |

## 4 EXPERIMENTS

### 4.1 EXPERIMENTAL SETUP

**Models and Datasets.** For natural language processing tasks, we use the pretrained Gemma-2B (Team et al., 2024) and LLaMA3-8B (Grattafiori et al., 2024). For commonsense reasoning tasks, we fine-tune on the Commonsense-170K dataset (Hu et al., 2023) and evaluate on eight standard benchmarks including BoolQ (Clark et al., 2019), PIQA (Bisk et al., 2020), SIQA (Sap et al., 2019), HellaSwag (Zellers et al., 2019), WinoGrande (Sakaguchi et al., 2020), ARC-c/e (Clark et al., 2018), and OBQA (Mihaylov et al., 2018). For natural language generation tasks, we select 100K examples from the MetaMathQA (Yu et al., 2024) and Code-Feedback (Zheng et al., 2024) datasets for fine-tuning. To assess performance, we use MATH (Hendrycks et al., 2021) and GSM8K (Cobbe et al., 2021) benchmarks for the models fine-tuned on MetaMathQA, and HumanEval (Chen et al., 2021) benchmark for the models fine-tuned on Code-Feedback. For the details on image classification tasks, see Section G.

**Implementation Details.** For optimization, we adopt the AdamW optimizer (Loshchilov & Hutter, 2019) following the standard setting. We set $\beta_1 = 0.9$, $\beta_2 = 0.999$, and apply zero weight decay for the optimizer. For all experiments, we set the hyperparameters of the proposed initialization to $N = 5$ and $T = 10$. For the scaling factor $s$, we fix it to 2 for LoRA and $\frac{\alpha}{\sqrt{r}}$ for the other methods, following rsLoRA (Kalajdzievski, 2023), where $\alpha$ is set to 4. We compare our method with the original LoRA and its recent variants, including rsLoRA (Kalajdzievski, 2023), LoRA+ (Hayou et al., 2024b) with scaling ratio 4, PiSSA (Meng et al., 2024), and DoRA (Liu et al., 2024). We reproduce the results of these methods under the same setting. We report the mean and standard deviation across three trials. Further details are provided in Table 9.

Table 4: **Ablation studies.** 'Init.' indicates the proposed initialization method, and $\text{Cap}_{\boldsymbol{X}}(\Delta \boldsymbol{W})$ and 'Optimal $\tilde{\boldsymbol{G}}$' denote the representational capacity of weight updates and the equivalent gradient achieving the optimal condition in Theorem 1.

| Method | $\text{Cap}_{\boldsymbol{X}}(\Delta \boldsymbol{W}) = \infty$ | Optimal $\tilde{G}$ | Init. | Gemma-2B | | | LLaMA3-8B | | |
|---|---|---|---|---|---|---|---|---|---|
| | | | | MATH | GSM8K | HumanEval | MATH | GSM8K | HumanEval |
| Full FT. | - | - | - | $19.17_{\pm 0.22}$ | $56.23_{\pm 0.23}$ | $33.35_{\pm 0.65}$ | $26.47_{\pm 0.41}$ | $77.04_{\pm 0.18}$ | $49.11_{\pm 0.44}$ |
| rsLoRA | ✓ | ✗ | ✗ | $18.09_{\pm 0.04}$ | $51.93_{\pm 0.53}$ | $31.10_{\pm 1.32}$ | $25.97_{\pm 0.17}$ | $76.28_{\pm 0.96}$ | $44.36_{\pm 1.39}$ |
| rsLoRA w/ Optimal $\tilde{G}$ | ✗ | ✓ | ✗ | $17.22_{\pm 0.24}$ | $51.18_{\pm 0.29}$ | $31.30_{\pm 0.99}$ | $24.14_{\pm 0.27}$ | $75.68_{\pm 0.35}$ | $45.04_{\pm 1.02}$ |
| OD-LoRA w/ $\text{Cap}_{\boldsymbol{X}}(\Delta \boldsymbol{W}) < \infty$ | ✗ | ✓ | ✓ | $17.28_{\pm 0.31}$ | $51.25_{\pm 0.29}$ | $31.71_{\pm 0.49}$ | $25.64_{\pm 0.22}$ | $76.27_{\pm 0.41}$ | $45.73_{\pm 0.86}$ |
| OD-LoRA w/ Suboptimal $\tilde{G}$ | ✓ | ✗ | ✓ | $18.18_{\pm 0.11}$ | $53.73_{\pm 0.19}$ | $32.32_{\pm 1.73}$ | $25.56_{\pm 0.40}$ | $76.02_{\pm 0.32}$ | $44.92_{\pm 0.29}$ |
| **OD-LoRA** | ✓ | ✓ | ✓ | $\mathbf{18.47}_{\pm 0.14}$ | $\mathbf{55.34}_{\pm 1.10}$ | $\mathbf{34.15}_{\pm 0.50}$ | $26.45_{\pm 0.39}$ | $77.22_{\pm 0.19}$ | $\mathbf{47.76}_{\pm 1.25}$ |

Figure 2: **Ablation studies using training loss curve (left) and alignment between the full gradient ($\boldsymbol{G}$) and the equivalent gradient ($\tilde{\boldsymbol{G}}$) (right).** Experiments are conducted using LLaMA3-8B and MetaMATHQA. We average the cosine similarity across all target layers.

## 4.2 COMMONSENSE REASONING

Table 1 shows the experimental results on the commonsense reasoning tasks. OD-LoRA outperforms LoRA and its variants in nearly all configurations, leading to the highest adaptation performance on average across all models and ranks. Specifically, OD-LoRA yields improvements of approximately 9 and 2 percentage points over LoRA on Gemma-2B and LLaMA3-8B, respectively, and surpasses recent variants including PiSSA and DoRA by 3–5 and 0.5–1 percentage points on Gemma-2B and LLaMA3-8B, respectively. In particular, Gemma-2B with OD-LoRA shows a substantial improvement over the others on the challenging HellaSwag benchmark, achieving about a 40 percentage point improvement over LoRA. Compared to the full fine-tuning (Full FT.), OD-LoRA with rank-32 achieves minimal performance gaps, demonstrating the effectiveness of OD-LoRA.

## 4.3 NATURAL LANGUAGE GENERATION

Table 2 shows the experimental results on the natural language generation tasks. OD-LoRA achieves the best adaptation performance on all benchmarks across all models and ranks. Specifically, OD-LoRA achieves improvements of approximately 1–2, 1–10, and 2–6 percentage points over existing methods on MATH, GSM8K, and HumanEval, respectively. Consistent with the commonsense reasoning results, OD-LoRA demonstrates substantial performance gains with Gemma-2B, highlighting its effectiveness, especially in challenging adaptation contexts. Furthermore, OD-LoRA with rank-32 achieves results comparable to or exceeding those of full fine-tuning. These results demonstrate that OD-LoRA significantly enhances the performance of LoRA.

## 4.4 IMAGE CLASSIFICATION

In addition to natural language processing tasks, we provide the results on image classification tasks. See Section G for details and more results. The results in Table 3 demonstrate that our method improves the performance of LoRA substantially, surpassing that of the other methods by significant margins. In particular, we observe that OD-LoRA shows about 2-3 percentage points higher accuracy in the Cars dataset, achieving the smallest performance gap compared to the full fine-tuning case. These results demonstrate the effectiveness of OD-LoRA in the vision domain.

## 4.5 ABLATION STUDIES AND CONVERGENCE ANALYSIS

To validate our claim on the dilemma in LoRA, we conduct ablation studies. Specifically, we propose 3 variants of rsLoRA and OD-LoRA: 'rsLoRA w/ Optimal $\tilde{G}$', 'OD-LoRA w/ $\text{Cap}_{\boldsymbol{X}}(\Delta \boldsymbol{W}) < \infty$' and 'OD-LoRA w/ Suboptimal $\tilde{G}$'. For the first and second methods, we enforce uniform singular values on the weight updates to satisfy the condition for accurate gradient approximation, which restricts their representational capacity. The second method, in addition, incorporates the proposed initialization phase. For the third method, we modify OD-LoRA by intentionally violating the uniform singular value condition for the equivalent gradient $\tilde{\mathcal{G}}$ in Theorem 3. See Section I for more details.

Table 5: **Comparison with existing gradient-based methods.** We reproduce the results of existing methods. Experiments are conducted with rank 8. See Table 7 for more results and analyses.

| Method | Gemma-2B | | | LLaMA3-8B | | |
|---|---|---|---|---|---|---|
| | MATH | GSM8K | HumanEval | MATH | GSM8K | HumanEval |
| LoRA-GA | $17.02_{\pm0.28}$ | $48.60_{\pm0.74}$ | $30.69_{\pm0.76}$ | $24.74_{\pm0.23}$ | $75.21_{\pm1.01}$ | $45.12_{\pm1.22}$ |
| LoRA-Pro | $16.52_{\pm0.14}$ | $44.58_{\pm0.55}$ | $28.66_{\pm0.99}$ | $23.90_{\pm0.44}$ | $72.93_{\pm0.17}$ | $40.24_{\pm0.99}$ |
| rsLoRA + ScaledAdamW | $16.64_{\pm0.31}$ | $45.56_{\pm0.53}$ | $26.83_{\pm0.99}$ | $24.00_{\pm0.17}$ | $73.39_{\pm0.20}$ | $43.29_{\pm1.15}$ |
| AltLoRA | $16.62_{\pm0.28}$ | $45.34_{\pm0.81}$ | $28.66_{\pm0.91}$ | $23.66_{\pm0.68}$ | $73.69_{\pm0.23}$ | $42.07_{\pm0.82}$ |
| **OD-LoRA** | $\mathbf{17.79}_{\pm0.18}$ | $\mathbf{52.08}_{\pm0.53}$ | $\mathbf{31.30}_{\pm0.76}$ | $\mathbf{25.89}_{\pm0.30}$ | $\mathbf{76.62}_{\pm0.34}$ | $\mathbf{47.15}_{\pm1.44}$ |

Table 6: **Analysis of training overhead.** We train the LLaMA3-8B model on a single NVIDIA H200 GPU for one epoch. For OD-LoRA, we separately report the time overhead incurred by the proposed initialization method (first term) and the subsequent training (second term).

| Method | MetaMATHQA | | Code-Feedback | |
|---|---|---|---|---|
| | Training Time (hours) | GPU-Memory (GB) | Training Time (hours) | GPU-Memory (GB) |
| LoRA | 1.61 | 128.54 | 1.72 | 128.80 |
| **OD-LoRA** | 0.03 + 1.67 (**+5.59%**) | 129.16 (**+0.48%**) | 0.03 + 1.80 (**+5.17%**) | 129.77 (**+0.75%**) |

Table 4 presents the adaptation performance, and Figure 2 presents the training loss curves and gradient approximation quality. We observe that satisfying the uniform singular value condition leads to better gradient approximation quality, substantiating Theorem 1. However, we find that satisfying only one of the two requirements (i.e., full representational capacity of weight updates or accurate gradient approximation) fails to improve performance and loss convergence, highlighting the negative impact of the dilemma between them. In contrast, we observe that OD-LoRA, which satisfies both requirements simultaneously, achieves better loss convergence and gradient approximation quality, resulting in improved adaptation performance.

## 4.6 COMPARISON WITH EXISTING GRADIENT-BASED METHODS

We compare OD-LoRA with existing methods that view LoRA from the low-rank gradient approximation perspective, including LoRA-GA (Wang et al., 2024), LoRA-Pro (Wang et al., 2025), ScaledAdamW (Zhang & Pilanci, 2024), and AltLoRA (Yu et al., 2025). We reproduce the results of these methods with the same experimental setup as detailed in Table 9. For a fair comparison, we implement LoRA-Pro without tracking the momentum of full fine-tuning gradients. Table 5 shows that OD-LoRA outperforms existing methods by a substantial margin. Particularly, we observe that adjusting gradients for LoRA matrices (LoRA-Pro, ScaledAdamW, AltLoRA) typically results in substantial performance degradation. As discussed in Section D, we argue that although these methods minimize the gap between the full fine-tuning gradient and its low-rank approximation, modifying the original gradients for LoRA matrices results in ineffective subspace learning, degrading the gradient approximation quality. Indeed, we observe that these methods exhibit worse loss convergence (Figure 3) and lower alignment between the full gradient and its low-rank approximation (Figure 4b). These results further demonstrate that OD-LoRA effectively improves gradient approximation quality, which explains its performance gain.

## 4.7 TRAINING OVERHEAD

To demonstrate the efficiency of OD-LoRA, we present the time and memory overhead incurred by OD-LoRA in Table 6. The results show that OD-LoRA introduces approximately 5% overhead in time and less than 1% overhead in GPU-memory. As noted previously, LoRA-Pro, which reduces the gradient approximation error via gradient adjustment, incurs approximately 10% time and 24% memory overhead. Compared to LoRA-Pro, our method achieves substantially higher efficiency, particularly in memory consumption. These results further highlight the effectiveness of OD-LoRA.

## 5 CONCLUSION

In this paper, we demonstrate that the low-rank adaptation (LoRA) faces a dilemma between the representational capacity of weight updates and accurate gradient approximation. The proposed method, OD-LoRA, overcomes this dilemma by decoupling the singular values of weight updates from the equivalent low-rank gradient, achieving the two requirements at the same time. The experimental results on various benchmarks and pretrained models demonstrate that OD-LoRA significantly improves the gradient approximation quality, leading to better loss convergence and adaptation performance than existing methods.

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

# APPENDIX

The contents of the Appendix are as follows:

## A  PROOF OF THEOREM 1

*Proof.* Let $\langle \cdot, \cdot \rangle_F$ and $|| \cdot ||_F$ denote the Frobenius inner product and Frobenius norm, respectively. Since $\langle \mathcal{G}, \tilde{\mathcal{G}} \rangle = \sum_{l=1}^{L} \langle \boldsymbol{G}^l, \tilde{\boldsymbol{G}}^l \rangle_F$, it is enough to prove that $\langle \boldsymbol{G}^l, \tilde{\boldsymbol{G}}^l \rangle_F$ is maximized if $\boldsymbol{A}^l$ and $\boldsymbol{B}^l$ have uniform singular values. For simplicity, we omit the subscript $l$. Using the property of the inner product, we rewrite $\langle \boldsymbol{G}, \tilde{\boldsymbol{G}} \rangle_F$ as follows:

$$\langle \boldsymbol{G}, \tilde{\boldsymbol{G}} \rangle_F = \langle \boldsymbol{G}, s^2(\boldsymbol{G}\boldsymbol{A}^\top \boldsymbol{A} + \boldsymbol{B}\boldsymbol{B}^\top \boldsymbol{G}) \rangle_F$$
$$= s^2(\langle \boldsymbol{G}, \boldsymbol{G}\boldsymbol{A}^\top \boldsymbol{A} \rangle_F + \langle \boldsymbol{G}, \boldsymbol{B}\boldsymbol{B}^\top \boldsymbol{G} \rangle_F). \tag{6}$$

Since the row space of $\boldsymbol{A}$ and the column space of $\boldsymbol{B}$ are given, we can define unique orthogonal projections onto those subspaces: $\boldsymbol{P}_{\mathcal{R}(\boldsymbol{A})} = \boldsymbol{A}^\top (\boldsymbol{A}\boldsymbol{A}^\top)^{-1} \boldsymbol{A}$ and $\boldsymbol{P}_{\mathcal{C}(\boldsymbol{B})} = \boldsymbol{B}(\boldsymbol{B}^\top \boldsymbol{B})^{-1} \boldsymbol{B}^\top$. This allows us to rewrite $\boldsymbol{G}$ as $\boldsymbol{G} = \boldsymbol{G}\boldsymbol{P}_{\mathcal{R}(\boldsymbol{A})} + \boldsymbol{G}(\boldsymbol{I} - \boldsymbol{P}_{\mathcal{R}(\boldsymbol{A})})$ or $\boldsymbol{G} = \boldsymbol{P}_{\mathcal{C}(\boldsymbol{B})}\boldsymbol{G} + (\boldsymbol{I} - \boldsymbol{P}_{\mathcal{C}(\boldsymbol{B})})\boldsymbol{G}$. Thus, we can rewrite each term in the numerator of the last equation of Equation 6 as

$$\langle \boldsymbol{G}, \boldsymbol{G}\boldsymbol{A}^\top \boldsymbol{A} \rangle_F = \langle \boldsymbol{G}\boldsymbol{P}_{\mathcal{R}(\boldsymbol{A})}, \boldsymbol{G}\boldsymbol{A}^\top \boldsymbol{A} \rangle_F + \langle \boldsymbol{G}(\boldsymbol{I} - \boldsymbol{P}_{\mathcal{R}(\boldsymbol{A})}), \boldsymbol{G}\boldsymbol{A}^\top \boldsymbol{A} \rangle_F,$$
$$\langle \boldsymbol{G}, \boldsymbol{B}\boldsymbol{B}^\top \boldsymbol{G} \rangle_F = \langle \boldsymbol{P}_{\mathcal{C}(\boldsymbol{B})}\boldsymbol{G}, \boldsymbol{B}\boldsymbol{B}^\top \boldsymbol{G} \rangle_F + \langle (\boldsymbol{I} - \boldsymbol{P}_{\mathcal{C}(\boldsymbol{B})})\boldsymbol{G}, \boldsymbol{B}\boldsymbol{B}^\top \boldsymbol{G} \rangle_F. \tag{7}$$

The second terms in Equation 7 become zero since

$$\langle \boldsymbol{G}(\boldsymbol{I} - \boldsymbol{P}_{\mathcal{R}(\boldsymbol{A})}), \boldsymbol{G}\boldsymbol{A}^\top \boldsymbol{A} \rangle_F = \mathrm{Tr}(\boldsymbol{G}(\boldsymbol{I} - \boldsymbol{P}_{\mathcal{R}(\boldsymbol{A})})\boldsymbol{A}^\top \boldsymbol{A}\boldsymbol{G}^\top)$$
$$= \mathrm{Tr}(\boldsymbol{G}(\boldsymbol{A}^\top \boldsymbol{A} - \boldsymbol{A}^\top \boldsymbol{A})\boldsymbol{G}^\top),$$
$$\langle (\boldsymbol{I} - \boldsymbol{P}_{\mathcal{C}(\boldsymbol{B})})\boldsymbol{G}, \boldsymbol{B}\boldsymbol{B}^\top \boldsymbol{G} \rangle_F = \mathrm{Tr}(\boldsymbol{G}^\top \boldsymbol{B}\boldsymbol{B}^\top (\boldsymbol{I} - \boldsymbol{P}_{\mathcal{C}(\boldsymbol{B})})\boldsymbol{G})$$
$$= \mathrm{Tr}(\boldsymbol{G}^\top (\boldsymbol{B}\boldsymbol{B}^\top - \boldsymbol{B}\boldsymbol{B}^\top)\boldsymbol{G}).$$

Thus, we rewrite $\langle \boldsymbol{G}, \tilde{\boldsymbol{G}} \rangle_F$ as follows:

$$\langle \boldsymbol{G}, \tilde{\boldsymbol{G}} \rangle_F = s^2(\langle \boldsymbol{G}\boldsymbol{P}_{\mathcal{R}(\boldsymbol{A})}, \boldsymbol{G}\boldsymbol{A}^\top \boldsymbol{A} \rangle_F + \langle \boldsymbol{P}_{\mathcal{C}(\boldsymbol{B})}\boldsymbol{G}, \boldsymbol{B}\boldsymbol{B}^\top \boldsymbol{G} \rangle_F).$$

By the Cauchy-Schwarz inequality, we obtain

$$\langle \boldsymbol{G}, \tilde{\boldsymbol{G}} \rangle_F \leq s^2(||\boldsymbol{G}\boldsymbol{P}_{\mathcal{R}(\boldsymbol{A})}||_F \cdot ||\boldsymbol{G}\boldsymbol{A}^\top \boldsymbol{A}||_F + ||\boldsymbol{P}_{\mathcal{C}(\boldsymbol{B})}\boldsymbol{G}||_F \cdot ||\boldsymbol{B}\boldsymbol{B}^\top \boldsymbol{G}||_F),$$

where the equality holds when $\boldsymbol{G}\boldsymbol{A}^\top \boldsymbol{A} = c\boldsymbol{G}\boldsymbol{P}_{\mathcal{R}(\boldsymbol{A})}$ and $\boldsymbol{B}\boldsymbol{B}^\top \boldsymbol{G} = d\boldsymbol{P}_{\mathcal{C}(\boldsymbol{B})}\boldsymbol{G}$ for some $c > 0$ and $d > 0$. The sufficient condition for arbitrary $\boldsymbol{G}$ is given by: $\boldsymbol{A}^\top \boldsymbol{A} = c\boldsymbol{P}_{\mathcal{R}(\boldsymbol{A})}$ and $\boldsymbol{B}\boldsymbol{B}^\top = d\boldsymbol{P}_{\mathcal{C}(\boldsymbol{B})}$. This leads to

$$\langle \boldsymbol{G}, \tilde{\boldsymbol{G}} \rangle_F \leq s^2(c||\boldsymbol{G}\boldsymbol{P}_{\mathcal{R}(\boldsymbol{A})}||_F^2 + d||\boldsymbol{P}_{\mathcal{C}(\boldsymbol{B})}\boldsymbol{G}||_F^2). \tag{8}$$

From the fixed energy assumption, we set $||\boldsymbol{A}||_F^2 = E_{\boldsymbol{A}}$ and $||\boldsymbol{B}||_F^2 = E_{\boldsymbol{B}}$. Since $\boldsymbol{A}$ and $\boldsymbol{B}$ have rank $r$, $\mathrm{Tr}(\boldsymbol{P}_{\mathcal{R}(\boldsymbol{A})}) = \mathrm{Tr}(\boldsymbol{P}_{\mathcal{C}(\boldsymbol{B})}) = r$. This gives $||\boldsymbol{A}||_F^2 = \mathrm{Tr}(\boldsymbol{A}^\top \boldsymbol{A}) = \mathrm{Tr}(c\boldsymbol{P}_{\mathcal{R}(\boldsymbol{A})}) = cr$ and $||\boldsymbol{B}||_F^2 = \mathrm{Tr}(\boldsymbol{B}\boldsymbol{B}^\top) = \mathrm{Tr}(d\boldsymbol{P}_{\mathcal{C}(\boldsymbol{B})}) = dr$, leading to $c = \frac{E_{\boldsymbol{A}}}{r}$ and $d = \frac{E_{\boldsymbol{B}}}{r}$. Therefore, we can rewrite Equation 8 as

$$\langle \boldsymbol{G}, \tilde{\boldsymbol{G}} \rangle_F \leq \frac{s^2}{r}(E_{\boldsymbol{A}}||\boldsymbol{G}\boldsymbol{P}_{\mathcal{R}(\boldsymbol{A})}||_F^2 + E_{\boldsymbol{B}}||\boldsymbol{P}_{\mathcal{C}(\boldsymbol{B})}\boldsymbol{G}||_F^2). \tag{9}$$

Since all the terms except for $G$ in Equation 9 are fixed, we can conclude that $\langle G, \tilde{G} \rangle_F$ is maximized for arbitrary $G$ if $A^\top A = \frac{E_A}{r} P_{\mathcal{R}(A)}$ and $BB^\top = \frac{E_B}{r} P_{\mathcal{C}(B)}$. Let $A = U_A \Sigma_A V_A^\top$ and $B = U_B \Sigma_B V_B^\top$ be the compact singular value decomposition of $A$ and $B$, respectively. By substituting $A$ and $B$ with their SVD, we obtain $A^\top A = V_A \Sigma_A^2 V_A^\top$, $BB^\top = U_B \Sigma_B^2 U_B^\top$, $P_{\mathcal{R}(A)} = V_A V_A^\top$, and $P_{\mathcal{C}(B)} = U_B U_B^\top$. From $A^\top A = \frac{E_A}{r} P_{\mathcal{R}(A)}$ and $BB^\top = \frac{E_B}{r} P_{\mathcal{C}(B)}$, we obtain

$$\Sigma_A^2 = \frac{E_A}{r} I_r,$$

$$\Sigma_B^2 = \frac{E_B}{r} I_r,$$

where $I_r$ is the $r \times r$ identity matrix. This implies that $A$ and $B$ have uniform singular values. $\qquad\square$

## B  PROOF OF THEOREM 2

Let $X \in \mathbb{R}^{d_{\text{out}} \times d_{\text{in}}}$ be an arbitrary rank-$r$ matrix with non-zero singular values $\{\sigma_1, \sigma_2, \ldots, \sigma_r\}$ and $\Delta W_{\text{LoRA}} = sBA$ where both $A$ and $B$ are rank-$r$. We prove the propositions in Theorem 2.

**Proof of "Without any constraints, $\text{Cap}_X(\Delta W_{\text{LoRA}}) = \infty$."**

*Proof.* Let $X = U_r \Sigma_r V_r^\top$ be the rank-$r$ truncated SVD of $X$. Then, we can always choose $A$ and $B$ such that $A = \Sigma_r V_r^\top$ and $B = \frac{1}{s} U_r$, resulting in $\min_{\Delta W_{\text{LoRA}}} ||X - \Delta W_{\text{LoRA}}||_F^2 = 0$. Thus, $\text{Cap}_X(\Delta W_{\text{LoRA}}) = \frac{1}{\min_{\Delta W_{\text{LoRA}}} ||X - \Delta W_{\text{LoRA}}||_F^2} = \infty$. $\qquad\square$

**Proof of "If $A$ and $B$ have uniform singular values, the non-zero singular values of $\Delta W_{\text{LoRA}}$ are uniform."**

*Proof.* We will examine the eigenvalues of $\Delta W_{\text{LoRA}}^\top \Delta W_{\text{LoRA}}$ since they are identical to the singular values of $\Delta W_{\text{LoRA}}$. The compact SVD of $B$ gives us $B = U_B \Sigma_B V_B^\top = \sigma_B U_B V_B^\top$. Then, we obtain $B^\top B = \sigma_B^2 V_B U_B^\top U_B V_B^\top = \sigma_B^2 I_r$ where $I_r$ denotes the $r \times r$ identity matrix. This gives us $\Delta W_{\text{LoRA}}^\top \Delta W_{\text{LoRA}} = s^2 A^\top B^\top B A = s^2 \sigma_B^2 A^\top A$, implying that the eigenvalues of $\Delta W_{\text{LoRA}}^\top \Delta W_{\text{LoRA}}$ are identical to those of $s^2 \sigma_B^2 A^\top A$. Since $A$ has uniform singular values, i.e., $\sigma_A$, the non-zero eigenvalues of $A^\top A$ are $\sigma_A^2$, which implies that the non-zero eigenvalues of $\Delta W_{\text{LoRA}}^\top \Delta W_{\text{LoRA}}$ are $s^2 \sigma_A^2 \sigma_B^2$. Therefore, the non-zero singular values of $\Delta W_{\text{LoRA}}$ are uniform, with value $s\sigma_A \sigma_B$. $\qquad\square$

**Proof of "If the non-zero singular values of $\Delta W_{\text{LoRA}}$ are uniform, $\text{Cap}_X(\Delta W_{\text{LoRA}}) = 1/\sum_{i=1}^r (\sigma_i - \frac{1}{r} \sum_{i=1}^r \sigma_i)^2$."**

*Proof.* Let $W$ denote $\Delta W_{\text{LoRA}}$ for simplicity. Recall $\text{Cap}_X(W) = 1/\min_W ||X - W||_F^2$. We can rewrite the Frobenius term in the denominator as

$$||X - W||_F^2 = \text{Tr}((X - W)^\top (X - W))$$
$$= \text{Tr}(X^\top X) - \text{Tr}(X^\top W) - \text{Tr}(W^\top X) + \text{Tr}(W^\top W)$$
$$= \text{Tr}(X^\top X) + \text{Tr}(W^\top W) - 2\text{Tr}(X^\top W).$$

Let $\sigma_W$ be the non-zero singular value of $W$. Then, we can evaluate the first and the second term: $\text{Tr}(X^\top X) = ||X||_F^2 = \sum_{i=1}^r \sigma_i^2$ and $\text{Tr}(W^\top W) = ||W||_F^2 = \sum_{i=1}^r \sigma_W^2 = r\sigma_W^2$. By the Von Neumann's Trace inequality, we obtain the following inequality for the third term:

$$\text{Tr}(X^\top W) \leq \sum_{i=1}^r \sigma_i \cdot \sigma_W,$$

where the equality holds if $X$ and $W$ share the same singular vectors. Thus, we obtain

$$||X - W||_F^2 \geq \sum_{i=1}^r \sigma_i^2 + r\sigma_W^2 - 2\sigma_W \sum_{i=1}^r \sigma_i.$$

As shown in this section, $W$ can represent any rank-$r$ matrix, implying that there exists $W$ that shares the same singular vectors with $X$. Thus, the minimization of $||X - W||_F^2$ is converted to the following optimization problem:

$$\min_{\sigma_W}(\sum_{i=1}^{r}\sigma_i^2 + r\sigma_W^2 - 2\sigma_W\sum_{i=1}^{r}\sigma_i),$$

which is a quadratic function of $\sigma_W$. Define $f(\sigma_W) = \sum_{i=1}^{r}\sigma_i^2 + r\sigma_W^2 - 2\sigma_W\sum_{i=1}^{r}\sigma_i$. By solving $\frac{df(\sigma_W)}{d\sigma_W} = 0$, we obtain $\sigma_W = \frac{1}{r}\sum_{i=1}^{r}\sigma_i = \bar{\sigma}$. Finally, this leads to $\text{Cap}_X(W) = 1/\min_W ||X - W||_F^2 = 1/f(\bar{\sigma}) = 1/\sum_{i=1}^{r}(\sigma_i - \bar{\sigma})^2$. $\qquad \square$

**Proof of "$\text{Cap}_X(\Delta W_{\text{LoRA}}) = \infty$ requires either $A$ or $B$ to have non-uniform singular values."**

*Proof.* Consider the contrapositive: *"If both $A$ and $B$ have uniform singular values, $\text{Cap}_X(\Delta W_{\text{LoRA}}) < \infty$."*. As shown in this section, if $A$ and $B$ have uniform singular values, the non-zero singular values of $\Delta W_{\text{LoRA}}$ are uniform, and if the non-zero singular values of $\Delta W_{\text{LoRA}}$ are uniform, $\text{Cap}_X(\Delta W_{\text{LoRA}}) = 1/\sum_{i=1}^{r}(\sigma_i - \frac{1}{r}\sum_{i=1}^{r}\sigma_i)^2$. Therefore, the contrapositive holds. $\quad \square$

## C  PROOF OF THEOREM 3

### C.1  PROOF OF THE FIRST PROPERTY

*Proof.* Since the gradient does not flow through $Q_A$ and $Q_B$, the gradient with respect to $A$ and $B$ is given by $\frac{\partial \mathcal{L}}{\partial A} = sQ_B^\top G$ and $\frac{\partial \mathcal{L}}{\partial B} = sGQ_A$. Then, the differential of $A$ and $B$ is expressed as

$$dA = -\eta\frac{\partial \mathcal{L}}{\partial A} = -\eta sQ_B^\top G,$$

$$dB = -\eta\frac{\partial \mathcal{L}}{\partial B} = -\eta sGQ_A.$$

Thus, we obtain

$$dW' = s(Q_B\,dA + dB\,Q_A^\top)$$
$$= -\eta s^2(GQ_AQ_A^\top + Q_BQ_B^\top G)$$
$$\tilde{G} = s^2(GQ_AQ_A^\top + Q_BQ_B^\top G).$$

$\qquad \square$

### C.2  PROOF OF THE SECOND PROPERTY

*Proof.* We have $A^\top = Q_AR_A$ and $B = Q_BR_B$. Then, we can rewrite $Q_BA + BQ_A^\top$ as follows:

$$Q_BA + BQ_A^\top = Q_BR_A^\top Q_A^\top + Q_BR_BQ_A^\top$$
$$= Q_B(R_A^\top + R_B)Q_A^\top.$$

For an arbitrary rank-$r$ matrix $X \in \mathbb{R}^{d_{\text{out}} \times d_{\text{in}}}$, let $X = U_r\Sigma_rV_r^\top$ be the rank-$r$ truncated SVD of $X$. Then, it is enough to prove that there exist $Q_A$, $Q_B$, $R_A$, and $R_B$ such that

$$Q_B(R_A^\top + R_B)Q_A^\top = U_r\Sigma_rV_r^\top. \tag{10}$$

By the property of QR decomposition, both $Q_A$ and $Q_B$ are matrices with orthonormal columns and both $R_A$ and $R_B$ are $r \times r$ invertible upper-triangle matrices. Thus, we can set $Q_A = V_r$, $Q_B = U_r$, $R_A = k\Sigma_r$, and $R_B = (1 - k)\Sigma_r$ for some $0 < k < 1$, achieving the equality in Equation 10. Therefore, $\Delta W_{\text{OD-LoRA}} = s(Q_BA + BQ_A^\top)$ can represent any rank-$r$ matrix $X$, leading to $\text{Cap}_X(\Delta W_{\text{OD-LoRA}}) = \infty$. $\qquad \square$

# D    COMPARISONS WITH RELATED WORKS

In this section, we provide a comprehensive comparison with existing methods that view LoRA from the perspective of low-rank gradient approximation (Zhang & Pilanci, 2024; Wang et al., 2025; Yu et al., 2025). While these methods share our goal of improving the alignment between the full fine-tuning gradient and its low-rank approximation, they rely on preconditioning the gradients of the LoRA matrices. In Section D.1, we theoretically analyze the difference between such preconditioning-based methods and OD-LoRA. Our analysis focuses on two key aspects: the effectiveness in subspace learning (Section D.1.1) and the resulting convergence rates under standard optimization assumptions (Section D.1.2). Finally, in Section D.2, we present empirical comparisons regarding loss convergence and gradient approximation quality to validate our theoretical findings.

## D.1    THEORETICAL ANALYSIS

In this subsection, we provide theoretical analyses of OD-LoRA and preconditioning-based methods under the SGD training setting. We adopt ScaledGD (Zhang & Pilanci, 2024) as a representative preconditioning-based method. Note that our analysis can be applied to the other methods (Wang et al., 2025; Yu et al., 2025) since they have similar update rules. For notational brevity, we present the analysis using single-layer notation because the generalization to the multi-layer setting is straightforward due to the additivity of the Frobenius norm and inner product. Let $\boldsymbol{\theta} = (\boldsymbol{A}, \boldsymbol{B})$ and $\boldsymbol{W}_{\mathrm{eff}} = \boldsymbol{W}_0 + \Delta \boldsymbol{W}(\boldsymbol{A}, \boldsymbol{B})$ denote the LoRA parameters and the effective weight, respectively. For simplicity, we ignore the scaling factor $s$ in our analysis. Also, let $\boldsymbol{G} = \nabla_{\boldsymbol{W}_{\mathrm{eff}}} \mathcal{L}$ denote the full fine-tuning gradient for $\boldsymbol{W}_{\mathrm{eff}}$. Let $\boldsymbol{P}_{\mathcal{R}(\boldsymbol{A})}$ and $\boldsymbol{P}_{\mathcal{C}(\boldsymbol{B})}$ denote the projection matrix onto the row space of $\boldsymbol{A}$ and the column space of $\boldsymbol{B}$, respectively. For a learning rate $\eta$, the update rule for OD-LoRA is given by

$$\Delta \boldsymbol{A} = -\eta \boldsymbol{Q}_B \boldsymbol{G}$$
$$\Delta \boldsymbol{B} = -\eta \boldsymbol{G} \boldsymbol{Q}_A \tag{11}$$
$$\Delta \boldsymbol{W}_{\mathrm{eff}} \approx -\eta(\boldsymbol{P}_{\mathcal{C}(\boldsymbol{B})} \boldsymbol{G} + \boldsymbol{G} \boldsymbol{P}_{\mathcal{R}(\boldsymbol{A})}).$$

The update rule for ScaledGD is given by:

$$\Delta \boldsymbol{A} = -\eta (\boldsymbol{B}^\top \boldsymbol{B})^{-1} \boldsymbol{B}^\top \boldsymbol{G}$$
$$\Delta \boldsymbol{B} = -\eta \boldsymbol{G} \boldsymbol{A}^\top (\boldsymbol{A} \boldsymbol{A}^\top)^{-1} \tag{12}$$
$$\Delta \boldsymbol{W}_{\mathrm{eff}} \approx -\eta(\boldsymbol{P}_{\mathcal{C}(\boldsymbol{B})} \boldsymbol{G} + \boldsymbol{G} \boldsymbol{P}_{\mathcal{R}(\boldsymbol{A})}).$$

Note that the update rule for $\boldsymbol{W}_{\mathrm{eff}}$ is identical in both methods. However, we demonstrate that the different update rules for $\boldsymbol{A}$ and $\boldsymbol{B}$ lead to different abilities to learn desirable subspaces (Section D.1.1), which in turn result in different convergence rates (Section D.1.2).

### D.1.1    SIGNAL-TO-NOISE RATIO ANALYSIS OF SUBSPACE LEARNING

We first demonstrate that OD-LoRA is effective in learning desirable subspaces for $\boldsymbol{A}$ and $\boldsymbol{B}$. In the context of LoRA, desirable subspaces have higher alignment with the full gradient, leading to a more accurate update of $\Delta \boldsymbol{W}_{\mathrm{eff}}$, as indicated in Equation 11 and Equation 12. Note that although the following analyses in this subsection focus on the gradient for $\boldsymbol{A}$, they are equally applicable to $\boldsymbol{B}$. Let $\boldsymbol{B} = \boldsymbol{U} \boldsymbol{\Sigma} \boldsymbol{V}^\top$ be the compact singular value decomposition for $\boldsymbol{B}$ where $\boldsymbol{U} = [\boldsymbol{u}_1, \boldsymbol{u}_2 \ldots, \boldsymbol{u}_r]$, $\boldsymbol{V} = [\boldsymbol{v}_1, \boldsymbol{v}_2 \ldots, \boldsymbol{v}_r]^\top$, and $\boldsymbol{\Sigma}$ is a diagonal matrix with singular values $\sigma_1 \geq \sigma_2 \geq \cdots \geq \sigma_r > 0$. To update $\boldsymbol{A}$, $\boldsymbol{u}_i$ is used to evaluate how the column space of $\boldsymbol{B}$ is aligned with the columns of $\boldsymbol{G}$. Then, the row space of $\boldsymbol{A}$ is updated according to the alignment. To evaluate how effectively the alignment information is used to update $\boldsymbol{A}$, we introduce the following definitions.

**Definition 2.** (**Update decomposition using alignment**) Express the update for $\boldsymbol{A}$ as a sum of components corresponding to the alignment between the left singular vectors of $\boldsymbol{B}$ and the full gradient $\boldsymbol{G}$:

$$\Delta \boldsymbol{A} = \sum_{i=i}^{r} \boldsymbol{x}_i (\boldsymbol{u}_i^\top \boldsymbol{G}) = \sum_{i=i}^{r} \boldsymbol{x}_i \boldsymbol{h}_i^\top, \tag{13}$$

- **Signal update** ($\boldsymbol{x}_1 \boldsymbol{h}_1^\top$): The component aligned with the strongest direction $\boldsymbol{u}_1$.

- **Noise update $(\boldsymbol{x}_r \boldsymbol{h}_r^\top)$:** The component aligned with the weakest direction $\boldsymbol{u}_r$.

**Definition 3.** (**Signal-to-noise ratio of update**) We define the signal-to-noise ratio of the update for $\boldsymbol{A}$ as

$$\text{SNR}(\Delta \boldsymbol{A}) := \frac{||\boldsymbol{x}_1 \boldsymbol{h}_1^\top||_F}{||\boldsymbol{x}_r \boldsymbol{h}_r^\top||_F}. \tag{14}$$

**Definition 4.** (**Condition number**) For a matrix $\boldsymbol{X}$, we define the condition number of $\boldsymbol{X}$ as

$$\kappa(\boldsymbol{X}) := \frac{\sigma_{\max}(\boldsymbol{X})}{\sigma_{\min}(\boldsymbol{X})}, \tag{15}$$

where $\sigma_{\max}(\boldsymbol{X})$ and $\sigma_{\min}(\boldsymbol{X})$ represent the largest and smallest singular value of $\boldsymbol{X}$.

For example, for the standard LoRA formulation, we can rewrite the update of $\boldsymbol{A}$ as $\Delta \boldsymbol{A} = -\eta \boldsymbol{B}^\top \boldsymbol{G} = \sum_{i=1}^{r} -\eta \sigma_i \boldsymbol{v}_i (\boldsymbol{u}_i^\top \boldsymbol{G}) = \sum_{i=1}^{r} -\eta \sigma_i \boldsymbol{v}_i \boldsymbol{h}_i^\top$. Then, the signal update is $-\eta \sigma_1 \boldsymbol{v}_1 \boldsymbol{h}_1^\top$ and the noise update is $-\eta \sigma_r \boldsymbol{v}_r \boldsymbol{h}_r^\top$. Consequently, the signal-to-noise ratio of the gradient for $\boldsymbol{A}$ in LoRA is expressed as $\frac{||-\eta \sigma_1 \boldsymbol{v}_1 \boldsymbol{h}_1^\top||_F}{||-\eta \sigma_r \boldsymbol{v}_r \boldsymbol{h}_r^\top||_F} = \kappa(\boldsymbol{B}) \frac{||\boldsymbol{h}_1||}{||\boldsymbol{h}_r||}$. This implies that the alignment information is distorted by $\kappa(\boldsymbol{B})$, causing strongly aligned directions to be amplified more.

Using these definitions, we derive the following theorem on the distortion of the signal-to-noise ratio:

---

**Theorem 4: Distortion of Signal-to-Noise Ratio**

*Let $SNR_{intrinsic} := \frac{||\boldsymbol{h}_1||}{||\boldsymbol{h}_r||}$ be the intrinsic signal-to-noise ratio of the gradient alignment. The Signal-to-Noise Ratio (SNR) of the parameter update $\Delta \boldsymbol{A}$ for OD-LoRA and ScaledGD is given by:*

- ***OD-LoRA:*** *$SNR(\Delta \boldsymbol{A}) = 1 \cdot SNR_{intrinsic}$.*
- ***Preconditioning-based Methods (e.g., ScaledGD):*** *$SNR(\Delta \boldsymbol{A}) = \frac{1}{\kappa(\boldsymbol{B})} SNR_{intrinsic}$.*

*This implies that existing methods amplify the noise component and suppress the signal component by the inverse of the singular values of $\boldsymbol{B}$, diluting the meaningful gradient signal.*

---

*Proof.* We analyze the update rules for each method using the decomposition in Definition 2.

**1. OD-LoRA**

The update rule is given by $\Delta \boldsymbol{A} = -\eta \boldsymbol{Q}_B^\top \boldsymbol{G}$. Since $\boldsymbol{Q}_B$ corresponds to the orthonormal basis $\boldsymbol{U}$ of the column space $\mathcal{C}(\boldsymbol{B})$, its singular values are all equal to $1$ ($\sigma_i = 1$ for all $i$). The update can be written as:

$$\Delta \boldsymbol{A} = \sum_{i=1}^{r} (-\eta \boldsymbol{v}_i)(\boldsymbol{u}_i^\top \boldsymbol{G}). \tag{16}$$

Here, the signal update is $-\eta \boldsymbol{v}_1 \boldsymbol{h}_1^\top$ and the noise update is $-\eta \boldsymbol{v}_r \boldsymbol{h}_r^\top$. Consequently, the SNR is calculated as:

$$\text{SNR}(\Delta \boldsymbol{A}) = \frac{||-\eta \boldsymbol{v}_1 \boldsymbol{h}_1^\top||_F}{||-\eta \boldsymbol{v}_r \boldsymbol{h}_r^\top||_F} = \frac{\eta ||\boldsymbol{h}_1||}{\eta ||\boldsymbol{h}_r||} = 1 \cdot \text{SNR}_{\text{intrinsic}}. \tag{17}$$

**2. ScaledGD**

The update rule is given by $\Delta \boldsymbol{A} = -\eta (\boldsymbol{B}^\top \boldsymbol{B})^{-1} \boldsymbol{B}^\top \boldsymbol{G}$. Using the SVD of $\boldsymbol{B} = \boldsymbol{U} \Sigma \boldsymbol{V}^\top$, the pseudoinverse term is $(\boldsymbol{B}^\top \boldsymbol{B})^{-1} \boldsymbol{B}^\top = \boldsymbol{V} \Sigma^{-1} \boldsymbol{U}^\top$. Substituting this into the update rule:

$$\Delta \boldsymbol{A} = -\eta \boldsymbol{V} \Sigma^{-1} \boldsymbol{U}^\top \boldsymbol{G} = \sum_{i=1}^{r} \left( -\eta \frac{1}{\sigma_i} \boldsymbol{v}_i \right) (\boldsymbol{u}_i^\top \boldsymbol{G}). \tag{18}$$

Here, the signal update is $-\frac{\eta}{\sigma_1} \boldsymbol{v}_1 \boldsymbol{h}_1^\top$, and the noise update is $-\frac{\eta}{\sigma_r} \boldsymbol{v}_r \boldsymbol{h}_r^\top$. Consequently, the SNR is calculated as:

$$\text{SNR}(\Delta \boldsymbol{A}) = \frac{||-\frac{\eta}{\sigma_1} \boldsymbol{v}_1 \boldsymbol{h}_1^\top||_F}{||-\frac{\eta}{\sigma_r} \boldsymbol{v}_r \boldsymbol{h}_r^\top||_F} = \frac{\frac{1}{\sigma_1} ||\boldsymbol{v}_1|| ||\boldsymbol{h}_1||}{\frac{1}{\sigma_r} ||\boldsymbol{v}_r|| ||\boldsymbol{h}_r||} = \frac{\sigma_r}{\sigma_1} \frac{||\boldsymbol{h}_1||}{||\boldsymbol{h}_r||} = \frac{1}{\kappa(\boldsymbol{B})} \text{SNR}_{\text{intrinsic}}. \tag{19}$$

$\square$

**Discussion.** This SNR analysis is particularly critical in the context of LoRA, where the trainable matrices $\boldsymbol{A}$ and $\boldsymbol{B}$ typically exhibit large condition numbers, driven by the intrinsic low-rank nature of task-specific features (see Figure 5 and Figure 6). In such regimes, the smallest singular vectors often encode noise or transient artifacts rather than meaningful data signals. Consequently, scaling the noise components by the inverse of their singular values may dilute the meaningful gradient signal, thereby resulting in ineffective subspace learning. By preserving the intrinsic signal-to-noise ratio, OD-LoRA ensures that the dominant signal component is effectively captured, facilitating effective subspace evolution even when the singular value distribution of the desired weight updates is highly skewed.

### D.1.2 CONVERGENCE ANALYSIS

Moreover, we compare OD-LoRA with ScaledGD from the perspective of convergence rate. We first introduce several assumptions used in our analysis.

**Assumption 1.** ($\beta$-*smootheness*) *The loss function $\mathcal{L}$ is $\beta$-smooth with respect to $\boldsymbol{W}_{\mathrm{eff}}$:*

$$\mathcal{L}(\boldsymbol{W}_{\mathrm{eff}} + \Delta\boldsymbol{W}_{\mathrm{eff}}) \leq \mathcal{L}(\boldsymbol{W}_{\mathrm{eff}}) + \underbrace{\langle \boldsymbol{G}, \Delta\boldsymbol{W}_{\mathrm{eff}} \rangle_F}_{Descent} + \underbrace{\frac{\beta}{2}||\Delta\boldsymbol{W}_{\mathrm{eff}}||_F^2}_{Penalty}. \quad (20)$$

This assumption is required to ensure the curvature of the optimization landscape is bounded. This allows us to derive a step size regime under which the first-order Taylor approximation remains valid, guaranteeing monotonic descent of the loss function.

**Assumption 2.** (*PL condition*) *The loss function $\mathcal{L}$ satisfies the $\mu$-Polyak–Łojasiewicz (PL) condition:* $||\nabla_{\boldsymbol{W}_{\mathrm{eff}}}\mathcal{L}||_F^2 \geq 2\mu(\mathcal{L}(\boldsymbol{W}_{\mathrm{eff}}) - \mathcal{L}^*)$ *where $\mathcal{L}^*$ denotes the minimal loss, which is assumed to be 0 under the overparameterized regime.*

Unlike strong convexity conditions, the PL condition allows for non-convex landscapes with multiple global minima, which is typical in deep learning. This condition relates the gradient magnitude to the sub-optimality gap, providing the necessary geometry to establish a linear convergence rate.

**Assumption 3.** ($\alpha$-*alignment*) *We assume that the projection of full gradient $\mathcal{G}$ onto $\mathcal{R}(\boldsymbol{A})$ and $\mathcal{C}(\boldsymbol{B})$ captures fraction $\alpha$ of the energy of $\mathcal{G}$ where $0 < \alpha \leq 2$:*

$$||\boldsymbol{G}\boldsymbol{P}_{\mathcal{R}(\boldsymbol{A})}||_F^2 + ||\boldsymbol{P}_{\mathcal{C}(\boldsymbol{B})}\boldsymbol{G}||_F^2 = \alpha||\boldsymbol{G}||_F^2. \quad (21)$$

In Equation 20, there are two terms to determine the loss-decrease bound: the descent term and the penalty term. We first derive lemmas on these terms.

**Lemma 1.** (*Descent term*) *For both OD-LoRA and ScaledGD, whose update rules are formulated as Equation 11 and Equation 12, respectively, the descent term in Equation 20 is given by*

$$\langle \boldsymbol{G}, \Delta\boldsymbol{W}_{\mathrm{eff}} \rangle_F = -\alpha\eta||\boldsymbol{G}||_F^2. \quad (22)$$

*Proof.* For both methods, the effective weight update is a projection of the gradient onto the low-rank subspaces: $\Delta\boldsymbol{W}_{\mathrm{eff}} \approx -\eta(\boldsymbol{P}_{\mathcal{C}(\boldsymbol{B})}\boldsymbol{G} + \boldsymbol{G}\boldsymbol{P}_{\mathcal{R}(\boldsymbol{A})})$. Substituting this into the inner product:

$$\begin{aligned}
\langle \boldsymbol{G}, \Delta\boldsymbol{W}_{\mathrm{eff}} \rangle_F &\approx \langle \boldsymbol{G}, -\eta(\boldsymbol{P}_{\mathcal{C}(\boldsymbol{B})}\boldsymbol{G} + \boldsymbol{G}\boldsymbol{P}_{\mathcal{R}(\boldsymbol{A})}) \rangle_F \\
&= -\eta(\langle \boldsymbol{G}, \boldsymbol{P}_{\mathcal{C}(\boldsymbol{B})}\boldsymbol{G} \rangle_F + \langle \boldsymbol{G}, \boldsymbol{G}\boldsymbol{P}_{\mathcal{R}(\boldsymbol{A})} \rangle_F).
\end{aligned} \quad (23)$$

For any matrix $\boldsymbol{X}$ and projection $\boldsymbol{P}$, $\langle \boldsymbol{X}, \boldsymbol{P}\boldsymbol{X} \rangle_F = \mathrm{Tr}(\boldsymbol{X}^\top\boldsymbol{P}\boldsymbol{X}) = \mathrm{Tr}(\boldsymbol{X}^\top\boldsymbol{P}^\top\boldsymbol{P}\boldsymbol{X}) = ||\boldsymbol{P}\boldsymbol{X}||_F^2$. Applying this to our terms:

$$\begin{aligned}
\langle \boldsymbol{G}, \boldsymbol{P}_{\mathcal{C}(\boldsymbol{B})}\boldsymbol{G} \rangle_F &= ||\boldsymbol{P}_{\mathcal{C}(\boldsymbol{B})}\boldsymbol{G}||_F^2, \\
\langle \boldsymbol{G}, \boldsymbol{G}\boldsymbol{P}_{\mathcal{R}(\boldsymbol{A})} \rangle_F &= \langle \boldsymbol{G}^\top, \boldsymbol{P}_{\mathcal{R}(\boldsymbol{A})}^\top\boldsymbol{G}^\top \rangle_F = ||\boldsymbol{G}\boldsymbol{P}_{\mathcal{R}(\boldsymbol{A})}||_F^2.
\end{aligned} \quad (24)$$

Finally, applying the $\alpha$-alignment assumption in Equation 21, we obtain

$$\begin{aligned}
\langle \boldsymbol{G}, \Delta\boldsymbol{W}_{\mathrm{eff}} \rangle_F &= -\eta(||\boldsymbol{G}\boldsymbol{P}_{\mathcal{R}(\boldsymbol{A})}||_F^2 + ||\boldsymbol{P}_{\mathcal{C}(\boldsymbol{B})}\boldsymbol{G}||_F^2) \\
&= -\alpha\eta||\boldsymbol{G}||_F^2.
\end{aligned} \quad (25)$$

$\square$

**Lemma 2.** *(**Penalty term**) For OD-LoRA, the bound for the penalty term in Equation 20 is given by*

$$\text{Penalty}_{\text{OD-LoRA}} \leq \alpha\beta\eta^2||\boldsymbol{G}||_F^2. \tag{26}$$

*For ScaledGD, the bound for the penalty term is given by*

$$\text{Penalty}_{\text{ScaledGD}} \leq \alpha\beta\kappa^2\eta^2||\boldsymbol{G}||_F^2, \tag{27}$$

*where $\kappa = \frac{\max(\sigma_{\max}(\boldsymbol{A}),\sigma_{\max}(\boldsymbol{B}))}{\min(\sigma_{\min}(\boldsymbol{A}),\sigma_{\min}(\boldsymbol{B}))}$.*

*Proof.* We bound the penalty term $\frac{\beta}{2}||\Delta\boldsymbol{W}_{\text{eff}}||_F^2$ by relating it to the parameter ($\boldsymbol{\theta} = (\boldsymbol{A}, \boldsymbol{B})$) updates. For any factorization $\boldsymbol{W}_{\text{eff}} \approx \boldsymbol{W}_0 + \boldsymbol{BA}$ (or involving $\boldsymbol{Q}$), the triangle inequality and sub-multiplicative property of norms give us

$$\begin{aligned}
||\Delta\boldsymbol{W}_{\text{eff}}||_F &= ||\boldsymbol{B}\Delta\boldsymbol{A} + \Delta\boldsymbol{BA}||_F \\
&\leq ||\boldsymbol{B}||_2||\Delta\boldsymbol{A}||_F + ||\Delta\boldsymbol{B}||_F||\boldsymbol{A}||_2 \\
&\leq \sigma_{\max}(\boldsymbol{A}, \boldsymbol{B})(||\Delta\boldsymbol{A}||_F + ||\Delta\boldsymbol{B}||_F),
\end{aligned} \tag{28}$$

where $\sigma_{\max}(\boldsymbol{A}, \boldsymbol{B}) = \max(||\boldsymbol{A}||_2, ||\boldsymbol{B}||_2)$. Using $(x + y)^2 \leq 2x^2 + 2y^2$, we obtain

$$||\Delta\boldsymbol{W}_{\text{eff}}||_F^2 \leq 2\sigma_{\max}^2(\boldsymbol{A}, \boldsymbol{B})(||\Delta\boldsymbol{A}||_F^2 + ||\Delta\boldsymbol{B}||_F^2). \tag{29}$$

**1. OD-LoRA**

$\boldsymbol{Q_B}$ and $\boldsymbol{Q_A^\top}$ have orthonormal columns/rows, so their spectral norms are exactly 1. Thus, $\sigma_{\max}(\boldsymbol{Q_A}, \boldsymbol{Q_B}) = 1$. Using the update rules $\Delta\boldsymbol{A} = -\eta\boldsymbol{Q_B^\top G}$ and $\Delta\boldsymbol{B} = -\eta\boldsymbol{GQ_A}$:

$$\begin{aligned}
||\Delta\boldsymbol{A}||_F^2 &= \eta^2||\boldsymbol{Q_B^\top G}||_F^2 = \eta^2\operatorname{Tr}(\boldsymbol{G^\top Q_B Q_B^\top G}) = \eta^2||\boldsymbol{P}_{\mathcal{C}(\boldsymbol{B})}\boldsymbol{G}||_F^2. \\
||\Delta\boldsymbol{B}||_F^2 &= \eta^2||\boldsymbol{GQ_A}||_F^2 = \eta^2\operatorname{Tr}(\boldsymbol{GQ_A Q_A^\top G^\top}) = \eta^2||\boldsymbol{GP}_{\mathcal{R}(\boldsymbol{A})}||_F^2.
\end{aligned} \tag{30}$$

Substituting these into Equation 29 with $\sigma_{\max} = 1$:

$$||\Delta\boldsymbol{W}_{\text{eff}}||_F^2 \leq 2 \cdot 1^2 \cdot \eta^2(||\boldsymbol{P}_{\mathcal{C}(\boldsymbol{B})}\boldsymbol{G}||_F^2 + ||\boldsymbol{GP}_{\mathcal{R}(\boldsymbol{A})}||_F^2). \tag{31}$$

Using the $\alpha$-alignment assumption ($||\boldsymbol{P}_{\mathcal{C}(\boldsymbol{B})}\boldsymbol{G}||_F^2 + ||\boldsymbol{GP}_{\mathcal{R}(\boldsymbol{A})}||_F^2 = \alpha||\boldsymbol{G}||_F^2$):

$$||\Delta\boldsymbol{W}_{\text{eff}}||_F^2 \leq 2\alpha\eta^2||\boldsymbol{G}||_F^2. \tag{32}$$

The penalty is therefore $\frac{\beta}{2}||\Delta\boldsymbol{W}_{\text{eff}}||_F^2 \leq \alpha\beta\eta^2||\boldsymbol{G}||_F^2$.

**2. ScaledGD**

The updates for $\boldsymbol{A}$ involve pseudoinverse $\boldsymbol{B}^\dagger = (\boldsymbol{B^\top B})^{-1}\boldsymbol{B^\top}$. From the update rules for $\boldsymbol{A}$ in Equation 12, we obtain:

$$\begin{aligned}
||\Delta\boldsymbol{A}||_F^2 &= \eta^2||\boldsymbol{B}^\dagger\boldsymbol{G}||_F^2 \\
&= \eta^2||\boldsymbol{B}^\dagger(\boldsymbol{P}_{\mathcal{C}(\boldsymbol{B})}\boldsymbol{G} + (\boldsymbol{I} - \boldsymbol{P}_{\mathcal{C}(\boldsymbol{B})})\boldsymbol{G})||_F^2 \\
&= \eta^2||\boldsymbol{B}^\dagger(\boldsymbol{P}_{\mathcal{C}(\boldsymbol{B})}\boldsymbol{G})||_F^2 \\
&\leq \eta^2||\boldsymbol{B}^\dagger||_2^2||\boldsymbol{P}_{\mathcal{C}(\boldsymbol{B})}\boldsymbol{G}||_F^2.
\end{aligned} \tag{33}$$

Since $||\boldsymbol{B}^\dagger||_2 = \frac{1}{\sigma_{\min}(\boldsymbol{B})}$, we have:

$$||\Delta\boldsymbol{A}||_F^2 \leq \eta^2\frac{1}{\sigma_{\min}^2(\boldsymbol{B})}||\boldsymbol{P}_{\mathcal{C}(\boldsymbol{B})}\boldsymbol{G}||_F^2. \tag{34}$$

Similarly, $||\Delta\boldsymbol{B}||_F^2 \leq \eta^2\frac{1}{\sigma_{\min}^2(\boldsymbol{A})}||\boldsymbol{GP}_{\mathcal{R}(\boldsymbol{A})}||_F^2$.

Substituting these into Equation 29:

$$\begin{aligned}
||\Delta\boldsymbol{W}_{\text{eff}}||_F^2 &\leq 2\sigma_{\max}^2(\boldsymbol{A}, \boldsymbol{B})\left(\frac{\eta^2}{\sigma_{\min}^2(\boldsymbol{B})}||\boldsymbol{P}_{\mathcal{C}(\boldsymbol{B})}\boldsymbol{G}||_F^2 + \frac{\eta^2}{\sigma_{\min}^2(\boldsymbol{A})}||\boldsymbol{GP}_{\mathcal{R}(\boldsymbol{A})}||_F^2\right) \\
&\leq 2\kappa^2\eta^2(||\boldsymbol{P}_{\mathcal{C}(\boldsymbol{B})}\boldsymbol{G}||_F^2 + ||\boldsymbol{GP}_{\mathcal{R}(\boldsymbol{A})}||_F^2) \\
&= 2\alpha\kappa^2\eta^2||\boldsymbol{G}||_F^2.
\end{aligned} \tag{35}$$

The penalty is $\frac{\beta}{2}||\Delta\boldsymbol{W}_{\text{eff}}||_F^2 \leq \alpha\beta\kappa^2\eta^2||\boldsymbol{G}||_F^2$. $\qquad\square$

Finally, using Lemma 1 and Lemma 2, we derive the following theorem on the convergence rate for OD-LoRA and ScaledGD:

---

**Theorem 5: Convergence Rate**

*The convergence rate for OD-LoRA is $\rho_{OD\text{-}LoRA} = \mathcal{O}(\frac{\mu\alpha}{\beta})$:*

$$\mathcal{L}_{t+1} \leq (1 - \frac{\mu\alpha}{2\beta})\mathcal{L}_t. \tag{36}$$

*The convergence rate for preconditioning-based methods (e.g., ScaledGD) is $\rho_{precond} = \mathcal{O}(\frac{\mu\alpha}{\beta\kappa^2})$:*

$$\mathcal{L}_{t+1} \leq (1 - \frac{\mu\alpha}{2\beta\kappa^2})\mathcal{L}_t, \tag{37}$$

*where $\kappa = \frac{\max(\sigma_{\max}(\boldsymbol{A}), \sigma_{\max}(\boldsymbol{B}))}{\min(\sigma_{\min}(\boldsymbol{A}), \sigma_{\min}(\boldsymbol{B}))}$.*

---

*Proof.* Let $\mathcal{L}_t = \mathcal{L}(\boldsymbol{W}_{\text{eff}})$ and $\mathcal{L}_{t+1} = \mathcal{L}(\boldsymbol{W}_{\text{eff}} + \Delta\boldsymbol{W}_{\text{eff}})$.

### 1. OD-LoRA

Substitute the Descent and Penalty lemmas into Equation 20:

$$\mathcal{L}_{t+1} \leq \mathcal{L}_t - \alpha\eta\|\boldsymbol{G}\|_F^2 + \alpha\beta\eta^2\|\boldsymbol{G}\|_F^2. \tag{38}$$

To find the optimal step size, we define $f(\eta) = -\alpha\eta + \alpha\beta\eta^2$ and solve $f'(\eta^*) = -\alpha + 2\alpha\beta\eta = 0$. This gives us $\eta^* = \frac{1}{2\beta}$. By substituting $\eta^*$ back into Equation 38, we obtain:

$$\mathcal{L}_{t+1} \leq \mathcal{L}_t - \frac{\alpha}{2\beta}\|\boldsymbol{G}\|_F^2 + \frac{\alpha}{4\beta}\|\boldsymbol{G}\|_F^2 = \mathcal{L}_t - \frac{\alpha}{4\beta}\|\boldsymbol{G}\|_F^2. \tag{39}$$

Applying the PL condition ($\|\boldsymbol{G}\|_F^2 \geq 2\mu\mathcal{L}_t$), we have

$$\begin{aligned}
\mathcal{L}_{t+1} &\leq \mathcal{L}_t - \frac{\alpha}{4\beta}(2\mu\mathcal{L}_t) \\
&= (1 - \frac{\mu\alpha}{2\beta})\mathcal{L}_t.
\end{aligned} \tag{40}$$

### 2. ScaledGD

Substitute the Descent and Penalty lemmas for ScaledGD:

$$\mathcal{L}_{t+1} \leq \mathcal{L}_t - \alpha\eta\|\boldsymbol{G}\|_F^2 + \alpha\beta\kappa^2\eta^2\|\boldsymbol{G}\|_F^2. \tag{41}$$

Similarly, we set $f(\eta) = -\alpha\eta + \alpha\beta\kappa^2\eta^2$ and solve $f'(\eta^*) = -\alpha + 2\alpha\beta\kappa^2\eta = 0$, obtaining $\eta^* = \frac{1}{2\beta\kappa^2}$. By substituting $\eta^*$ into Equation 41:

$$\mathcal{L}_{t+1} \leq \mathcal{L}_t - \frac{\alpha}{2\beta\kappa^2}\|\boldsymbol{G}\|_F^2 + \frac{\alpha}{4\beta\kappa^2}\|\boldsymbol{G}\|_F^2 = \mathcal{L}_t - \frac{\alpha}{4\beta\kappa^2}\|\boldsymbol{G}\|_F^2. \tag{42}$$

Applying the PL condition:

$$\begin{aligned}
\mathcal{L}_{t+1} &\leq \mathcal{L}_t - \frac{\alpha}{4\beta\kappa^2}(2\mu\mathcal{L}_t) \\
&\leq (1 - \frac{\mu\alpha}{2\beta\kappa^2})\mathcal{L}_t.
\end{aligned} \tag{43}$$

$\square$

**Discussion.** Theorem 5 reveals that while both ScaledGD and OD-LoRA achieve geometric optimality by ensuring the effective weight update is an isotropic projection of the full gradient, they differ significantly in their stability and resulting convergence speeds. ScaledGD achieves this alignment through preconditioning, a mechanism that inherently scales parameter updates by the inverse of the smallest singular values. To mitigate the resulting parameter instability, the step size must be dampened by the square of the condition number ($\kappa^2$), as evidenced by the derived optimal step size $\eta^*$. In contrast, OD-LoRA resolves this issue by restructuring the parameterization via orthonormal bases, effectively enforcing a perfect condition number ($\kappa = 1$) by construction. Consequently, OD-LoRA is the only method that simultaneously secures the optimal descent direction and supports a large, constant learning rate, achieving a convergence rate independent of the weight matrix conditioning.

Table 7: **Comparison with existing gradient-based methods.** We reproduce the results of existing methods. Experiments are conducted with rank 32.

| Method | Gemma-2B | | | LLaMA3-8B | | |
|---|---|---|---|---|---|---|
| | MATH | GSM8K | HumanEval | MATH | GSM8K | HumanEval |
| LoRA-GA | $17.92_{\pm0.22}$ | $52.56_{\pm0.44}$ | $32.52_{\pm0.76}$ | $25.64_{\pm0.63}$ | $76.43_{\pm0.92}$ | $45.12_{\pm1.22}$ |
| LoRA-Pro | $17.32_{\pm0.41}$ | $52.11_{\pm0.25}$ | $30.49_{\pm0.76}$ | $25.02_{\pm0.44}$ | $76.02_{\pm0.37}$ | $43.90_{\pm0.82}$ |
| rsLoRA + ScaledAdamW | $17.22_{\pm0.42}$ | $52.37_{\pm0.63}$ | $31.10_{\pm1.22}$ | $25.21_{\pm0.17}$ | $75.49_{\pm0.28}$ | $44.76_{\pm0.86}$ |
| AltLoRA | $17.03_{\pm0.19}$ | $52.44_{\pm0.90}$ | $30.46_{\pm0.4}$ | $25.17_{\pm0.48}$ | $75.92_{\pm0.30}$ | $43.97_{\pm0.61}$ |
| **OD-LoRA** | $\mathbf{18.47}_{\pm0.14}$ | $\mathbf{55.34}_{\pm1.10}$ | $\mathbf{34.15}_{\pm0.50}$ | $\mathbf{26.45}_{\pm0.39}$ | $\mathbf{77.22}_{\pm0.19}$ | $\mathbf{47.76}_{\pm1.25}$ |

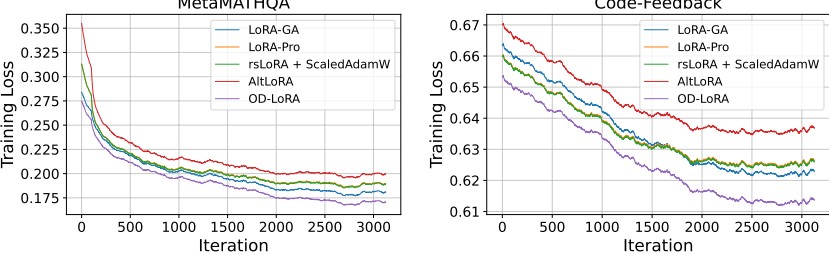

Figure 3: **Comparison with existing gradient-based methods using training loss curve.** Experiments are conducted on LLaMA3-8B with rank 8. We smooth the curves for better visualization.

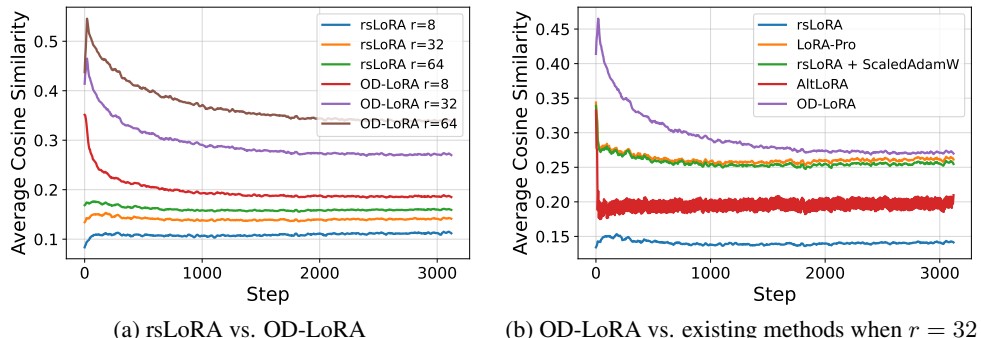

(a) rsLoRA vs. OD-LoRA      (b) OD-LoRA vs. existing methods when $r = 32$

Figure 4: **Cosine similarity between full fine-tuning gradient ($G$) and the equivalent gradient ($\tilde{G}$).** We measure the average cosine similarity across all target layers and present the rolling window mean over training steps, with a window size of 5.

## D.2 EMPIRICAL COMPARISONS

In this subsection, we provide empirical comparisons with existing gradient-based methods (Wang et al., 2024; 2025; Zhang & Pilanci, 2024; Yu et al., 2025). Table 7 shows the adaptation performance comparisons under the rank-32 setting. Similar to the rank-8 results in Table 5, the results demonstrate that OD-LoRA significantly outperforms existing methods. We also compare these methods using training loss curves. Figure 3 shows that OD-LoRA achieves faster convergence toward lower loss. Moreover, we provide the comparisons in terms of alignment between the full fine-tuning gradient and its low-rank approximation in each method. Figure 4b demonstrates that OD-LoRA achieves better gradient alignment. These empirical findings align with our theoretical analyses in Section D.1, which demonstrate that preconditioning in existing methods results in slow convergence due to the condition-number penalty ($\kappa^2$) and degraded gradient alignment caused by noise-amplified subspace learning. By decoupling the update magnitude from singular values, OD-LoRA avoids these pitfalls, resulting in the superior convergence speed and higher gradient alignment observed in our experiments.

## E    Importance of Improving Gradient Approximation Quality

In this section, we discuss the reason why improving gradient approximation quality is essential. Using the notations and assumptions introduced in Section 3.2, the descent lemma (Nesterov, 2018) gives us

$$\mathcal{L}(\mathcal{W} + \Delta\mathcal{W}) \leq \mathcal{L}(\mathcal{W}) + \langle \mathcal{G}, \Delta\mathcal{W} \rangle + \frac{\beta}{2}||\Delta\mathcal{W}||^2. \tag{44}$$

Equation 3 is simply derived if we set $\Delta\mathcal{W} = -\eta\tilde{\mathcal{G}}$, which implies updating $\mathcal{W}$ using alternative gradient $\tilde{\mathcal{G}}$. Equation 3 suggests that as the alternative gradient exhibits higher alignment with the full fine-tuning gradient, a steeper decrease of loss will be guaranteed. For empirical validation, we measure the cosine similarity between the full fine-tuning gradient and the equivalent gradient during training with rsLoRA or OD-LoRA. We adopt rsLoRA instead of LoRA to compare them under the same scaling factor, and we measure cosine similarity instead of inner product to consider the varying magnitude of the full gradient for each experiment. Figure 4a shows that as the rank of rsLoRA increases, the equivalent gradient exhibits higher alignment with the full gradient. Considering the better performance of LoRA with a higher rank, these results demonstrate the positive correlation between the adaptation performance of rsLoRA and the gradient approximation quality. Moreover, we observe that OD-LoRA significantly improves the alignment, suggesting that the performance gain of OD-LoRA is related to the better gradient alignment.

## F    Singular Value Distribution

In the main manuscript, we argue that desired weight updates may not favor uniform singular values. To justify this, we present the distribution of singular values learned by full fine-tuning. Figure 5 shows the singular value distribution in full fine-tuning across various datasets and modules. The results demonstrate that full fine-tuning tends to produce highly skewed singular values. Moreover, in Figure 6, we show the singular value distribution of weight updates learned via LoRA. Similar to the results for full fine-tuning, LoRA typically learns weight updates with non-uniform singular values.

## G    Experiments on Image Classification Tasks

We use ViT-Base and ViT-Large (Dosovitskiy et al., 2021) trained with 224×224 images and 16×16 patch size. We fine-tune and evaluate them on five datasets, including Cars (Krause et al., 2013), CUB200 (Wah et al., 2011), DTD (Cimpoi et al., 2014), Food101 (Bossard et al., 2014), and SUN397 (Xiao et al., 2010). We use the same setup as for the natural language processing tasks, except for the epochs, learning rate, and batch size, which are detailed in Table 10. We present the complete results in Table 8. The results show that OD-LoRA achieves the best average performance across all models and ranks. In particular, we observe that OD-LoRA significantly reduces the performance gap between LoRA and the full fine-tuning, achieving performance comparable to the full fine-tuning. These results suggest that OD-LoRA generalizes well to vision tasks, highlighting the effectiveness of our method in various domains.

## H    Additional Ablation Studies

In this section, we provide additional ablation studies on the hyperparameters of the proposed initialization method. In Algorithm 1, $N$ and $T$ denote the number of updates and the update interval during the initialization phase. We investigate how sensitive OD-LoRA's performance is to the choice of $N$ and $T$. Figure 7 shows OD-LoRA's performance across various configurations of $N$ and $T$. The results indicate that performance generally increases as both $N$ and $T$ increase. However, the overall differences are minor, suggesting that OD-LoRA is not sensitive to these hyperparameters.

## I    Details on Ablation Studies

In Section 4.5, we propose three variants of rsLoRA and OD-LoRA for ablation studies: 'rsLoRA w/ Optimal $\tilde{G}$', 'OD-LoRA w/ $\text{Cap}_X(\Delta W) < \infty$' and 'OD-LoRA w/ Suboptimal $\tilde{G}$'. The first

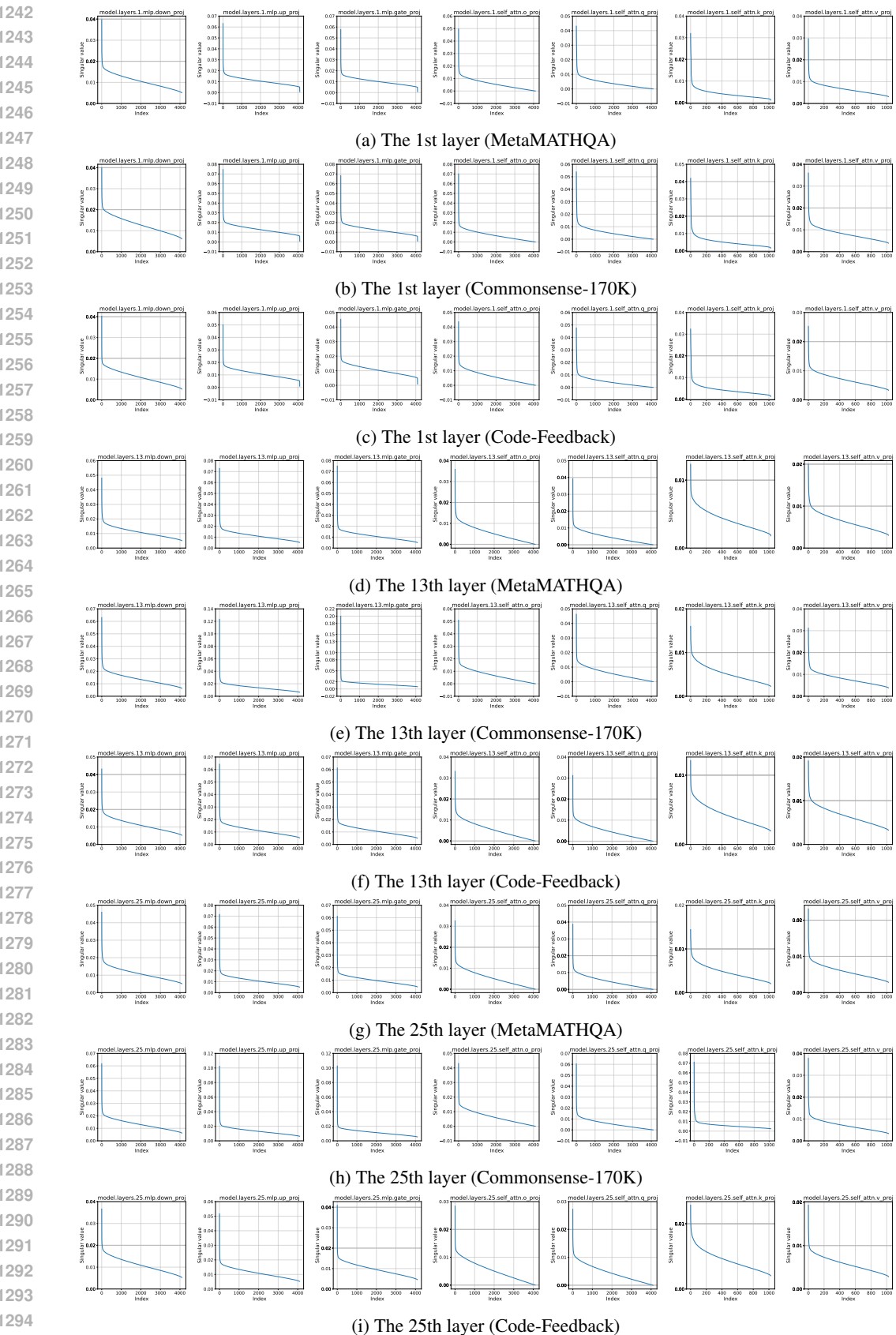

Figure 5: **Singular value distributions trained on LLaMA3-8B using full fine-tuning.** Best viewed in enlarged form.

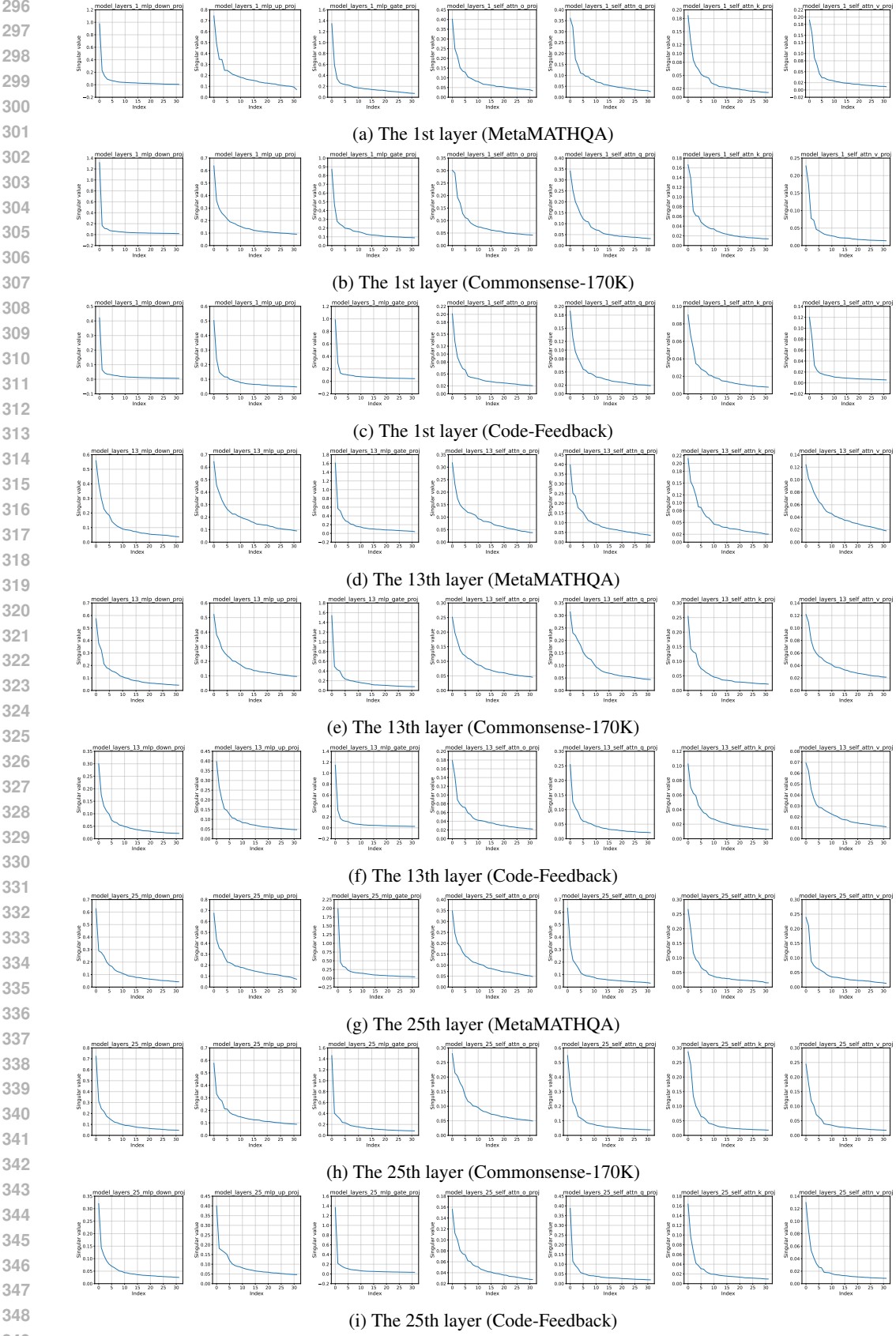

Figure 6: **Singular value distributions trained on LLaMA3-8B using LoRA with rank 32.** Best viewed in enlarged form.

Table 8: **Results on image classification tasks.**

| Model | Method | Rank | Cars | CUB200 | DTD | Food101 | SUN397 | Avg. |
|---|---|---|---|---|---|---|---|---|
| ViT-Base | Full FT. | - | 84.26±0.19 | 86.35±0.18 | 80.07±0.32 | 89.27±0.20 | 74.76±0.26 | 82.94 |
| | LoRA | 8 | 78.66±0.23 | 85.65±0.05 | 78.81±0.95 | 88.15±0.07 | 74.35±0.28 | 81.12 |
| | rsLoRA | | 78.48±0.07 | 85.67±0.41 | 78.79±0.48 | 88.13±0.16 | 74.63±0.13 | 81.14 |
| | LoRA+ | | 79.05±0.25 | 85.28±0.11 | 78.32±0.16 | 88.27±0.05 | 72.87±0.28 | 80.76 |
| | PiSSA | | 78.03±0.32 | 85.17±0.24 | 78.33±0.46 | 87.65±0.07 | 72.82±0.10 | 80.40 |
| | DoRA | | 78.93±0.30 | 85.74±0.35 | 78.46±0.16 | 88.27±0.17 | 74.46±0.09 | 81.17 |
| | **OD-LoRA** | | **80.47±0.14** | **86.08±0.16** | **79.36±0.30** | **88.35±0.11** | **74.92±0.16** | **81.84** |
| | LoRA | 32 | 81.48±0.20 | 85.75±0.28 | 79.34±0.18 | 88.37±0.12 | 70.82±0.07 | 81.15 |
| | rsLoRA | | 80.57±0.22 | 85.42±0.41 | 79.50±0.09 | 88.81±0.10 | 73.92±0.26 | 81.64 |
| | LoRA+ | | 79.53±0.60 | 85.68±0.31 | 78.85±0.39 | 88.05±0.01 | 67.66±0.06 | 79.95 |
| | PiSSA | | 80.43±0.17 | 84.90±0.28 | 78.90±0.35 | 87.91±0.08 | 69.73±0.12 | 80.37 |
| | DoRA | | 81.07±0.03 | 85.67±0.22 | 79.56±0.43 | 88.83±0.08 | 74.05±0.21 | 81.84 |
| | **OD-LoRA** | | **83.04±0.32** | **86.06±0.24** | **80.11±0.16** | **89.05±0.08** | **74.92±0.21** | **82.64** |
| ViT-Large | Full FT. | - | 88.19±0.13 | 88.23±0.19 | 81.71±0.09 | 90.90±0.17 | 76.20±0.22 | 85.05 |
| | LoRA | 8 | 85.07±0.19 | 87.53±0.20 | 80.64±0.23 | 89.96±0.08 | 76.30±0.11 | 83.90 |
| | rsLoRA | | 84.61±0.11 | 87.75±0.19 | 80.87±0.32 | 89.92±0.05 | 76.45±0.16 | 83.92 |
| | LoRA+ | | 81.99±0.97 | 87.66±0.05 | 80.62±0.55 | 90.06±0.03 | 75.08±0.14 | 83.08 |
| | PiSSA | | 85.59±0.17 | 87.55±0.19 | 80.34±0.13 | 89.60±0.17 | 75.41±0.20 | 83.70 |
| | DoRA | | 84.67±0.08 | 87.58±0.11 | 80.73±0.33 | 90.03±0.04 | 76.50±0.08 | 83.90 |
| | **OD-LoRA** | | **85.80±0.08** | **87.98±0.08** | **81.06±0.49** | **90.05±0.05** | **76.65±0.13** | **84.31** |
| | LoRA | 32 | 86.23±0.05 | 87.94±0.22 | 81.15±0.32 | 90.62±0.03 | 74.59±0.25 | 84.11 |
| | rsLoRA | | 85.93±0.30 | 87.64±0.06 | 81.56±0.07 | 90.58±0.13 | 76.22±0.13 | 84.39 |
| | LoRA+ | | 82.40±1.84 | 87.92±0.22 | 80.92±0.18 | 90.48±0.07 | 75.19±1.23 | 83.38 |
| | PiSSA | | 86.70±0.15 | 87.65±0.15 | 80.82±0.26 | 90.04±0.09 | 73.84±0.08 | 83.81 |
| | DoRA | | 86.10±0.36 | 87.69±0.11 | 81.31±0.13 | 90.64±0.04 | 76.23±0.20 | 84.39 |
| | **OD-LoRA** | | **87.07±0.36** | **88.27±0.10** | **81.45±0.21** | **90.75±0.17** | **76.48±0.06** | **84.80** |

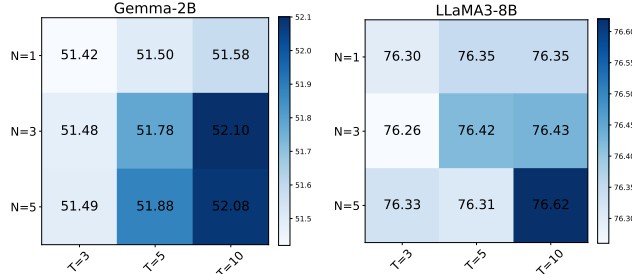

Figure 7: **Ablation studies on $N$ and $T$.** Using OD-LoRA, we fine-tune on MetaMATHQA and evaluate on GSM8K.

Table 9: **Additional implementation details.**

| | Full FT. | | LoRA-based | | |
|---|---|---|---|---|---|
| | Gemma-2B | LLaMA3-8B | Gemma-2B | LLaMA2-7B | LLaMA3-8B |
| Learning Rate | 2e-5 | 1e-5 | 2e-4 | 2e-4 | 1e-4 |
| Learning Rate Scheduler | | | cosine scheduler | | |
| Epochs | | | 1 | | |
| Batch Size | | | 32 | | |
| Target Modules | | 'q_proj', 'k_proj', 'v_proj', 'up_proj', 'down_proj', 'o_proj', 'gate_proj' | | | |

Table 10: **Implementation details on image classification tasks.**

| | ViT-Base | | | | | ViT-Large | | | | |
|---|---|---|---|---|---|---|---|---|---|---|
| | Cars | CUB200 | DTD | Food101 | SUN397 | Cars | CUB200 | DTD | Food101 | SUN397 |
| Epochs | 5 | 7 | 7 | 7 | 5 | 5 | 7 | 7 | 7 | 5 |
| Learning Rate | 5e-3 | 2e-3 | 2e-3 | 2e-3 | 5e-3 | 2.5e-3 | 1e-3 | 1e-3 | 1e-3 | 2.5e-3 |
| Batch Size | | | | | 64 | | | | | |
| Target Modules | | | | | 'query', 'value' | | | | | |

and second methods formulate the weight updates as $s\boldsymbol{Q_B}\boldsymbol{Q_A}^\top$. The third method formulates the weight updates as $s(\boldsymbol{Q_B}\mathrm{diag}(\boldsymbol{R_B})\boldsymbol{A} + \boldsymbol{B}(\boldsymbol{Q_A}\mathrm{diag}(\boldsymbol{R_A}))^\top)$. Except for the first method, $\boldsymbol{Q_A}$ and $\boldsymbol{Q_B}$ are initialized the same way as in OD-LoRAfor a fair comparison. The update interval for $\boldsymbol{Q_A}$ and $\boldsymbol{Q_B}$ is also the same as in OD-LoRA. In both 'rsLoRA w/ Optimal $\tilde{G}$' and 'OD-LoRA w/ $\mathrm{Cap}_{\boldsymbol{X}}(\Delta\boldsymbol{W}) < \infty$', $\boldsymbol{Q_A}$ and $\boldsymbol{Q_B}$ are trainable, but the gradient does not flow through the QR

decomposition process. Thus, the gradient for $Q_A$ and $Q_B$ are expressed as $\frac{\partial \mathcal{L}}{\partial Q_A^\top} = sQ_B^\top G$ and $\frac{\partial \mathcal{L}}{\partial Q_B} = sGQ_A$, leading to $\tilde{G} = s^2(GQ_AQ_A^\top + Q_BQ_B^\top G)$. For 'OD-LoRA w/ Suboptimal $\tilde{G}$', the gradient for $A$ and $B$ are expressed as $\frac{\partial \mathcal{L}}{\partial A} = s(Q_B \text{diag}(R_B))^\top G$ and $\frac{\partial \mathcal{L}}{\partial B} = sGQ_A \text{diag}(R_A)$, leading to $\tilde{G} = s^2(GQ_A \text{diag}(R_A)^2 Q_A^\top + Q_B \text{diag}(R_B)^2 Q_B^\top G)$. Hence, 'rsLoRA w/ Optimal $\tilde{G}$' and 'OD-LoRA w/ $\text{Cap}_X(\Delta W) < \infty$' satisfies the uniform singular value condition in Theorem 1, whereas 'OD-LoRA w/ Suboptimal $\tilde{G}$' does not. Moreover, the representational capacity of 'rsLoRA w/ Optimal $\tilde{G}$' and 'OD-LoRA w/ $\text{Cap}_X(\Delta W) < \infty$' is less than infinity, and that of 'OD-LoRA w/ Suboptimal $\tilde{G}$' is infinity as proven in Section B.

**Use of Large Language Models.** We use Large Language Models (LLMs) to assist with writing and to search for mathematical propositions, such as properties in linear algebra. We acknowledge our responsibility for the content of this paper and ensure that the contribution of LLMs is insignificant.

