# OpenReview forum: "OD-LoRA: Overcoming the Dilemma between Weight Representation and Gradient Approximation in Low-Rank Adaptation"
_ICLR.cc/2026/Conference — Submitted to ICLR 2026_

### Official Review · Reviewer_9ab4 · 2025-10-27

**Soundness:** 2
**Presentation:** 3
**Contribution:** 2
**Rating:** 4
**Confidence:** 4

**Summary:**

The paper identifies a theoretical dilemma in LoRA between weight update representation and gradient approximation. To address this, the authors propose OD-LoRA, which decouples the approximated gradient from the singular values of low-rank matrices, enabling accurate gradient approximation while preserving full representational capacity. The paper includes theoretical analysis and comprehensive experiments.

**Strengths:**

- The paper provides a novel and insightful perspective on LoRA by analyzing the alignment between the full gradient and low-rank subspaces.

 - The presentation is clear and logically structured, making the theoretical development easy to follow.

 - The experimental evaluation is comprehensive, covering multiple model sizes and domains, and demonstrates consistent improvement over baselines.

**Weaknesses:**

- W1. The paper argues that uniform singular values benefit gradient approximation, but this condition is not strictly necessary for minimizing angular distance. Prior works ([1] and [2]) on gradient projection in LoRA suggest the scaled gradient descent can project the full gradient onto low-rank spaces perfectly without enforcing uniform singular values. The paper also fails to compare with these key references, making its claimed novelty somewhat suspicious. The authors should clearly justify the necessity of this condition and clarify how OD-LoRA fundamentally differs from existing gradient-projection formulations.

- W2.  The statement (Line 266) that low-rank adapters must be initialized to zero is not valid. Several methods (e.g., LoRA-GA) adopt non-zero initialization by modifying pretrained parameters. Moreover, OD-LoRA comes to be the standard LoRA for the first few steps, which does not address gradient misalignment during early training. It would be valuable to discuss integrating the uniform singular value condition with alternative optimizers (e.g., Muon) and contrast against AdamW.


- W3. The algorithm is theory-inspired, but the paper lacks formal guarantees (e.g., convergence). Even a modest convergence statement or stability analysis would strengthen the theory.

- W4. Gradient-based baselines (LoRA-GA, LoRA-Pro) are compared on LLaMA2-7B, while the main experiments use LLaMA3-8B. Including these baselines on LLaMA3-8B (Instruct) would make the empirical case more convincing.


[1] Zhang F, Pilanci M. Riemannian preconditioned lora for fine-tuning foundation models[J]. arXiv preprint arXiv:2402.02347, 2024.

[2] Yu X, Wang Y, Chen J, et al. AltLoRA: Towards Better Gradient Approximation in Low-Rank Adaptation with Alternating Projections[J]. arXiv preprint arXiv:2505.12455, 2025.

**Questions:**

- Q1. The analysis focuses on the gradient alignment but ignores the effect of optimizer momentum (e.g., AdamW). Since momentum changes the effective update direction, how would this affect the theoretical claims?


- Q2. Typo error: Line 314: "trauncated" $\rightarrow$ "truncated".

---

> ### Author Response · Authors · 2025-11-24
>
> We appreciate your time and effort in reviewing our work. We are pleased to receive your positive comments regarding our novel and insightful perspective, the clear and logical structure, and the comprehensive experimental results. We also appreciate your pointing out several works that are closely related to ours. We hope that our responses adequately address the concerns you raised.
>
> ## **1. Missing related works and the difference from them (W1)**
>
> We confirmed that these works achieve the optimal condition in our Theorem 1 by modifying the gradient for the LoRA parameters $A$ and $B$. However, we argue that the main difference lies in how this optimal gradient approximation is achieved. **While previous methods manually adjust the gradient for LoRA parameters (preconditioning), the proposed new formulation of low-rank weight updates achieves this without gradient adjustment.** In Section D of the revised version, we discuss the potential negative impact of adjusting the original gradients from the perspective of the loss-decrease guarantee. Specifically, Equation 20 suggests that the loss-decrease guarantee deteriorates as the alternative gradients for updates deviate from the original. Similarly, the authors of LoRA-Pro [1] propose minimizing the difference between the adjusted gradient and the original, since a large difference results in significant performance degradation (see Theorem 2.3 and Table 6 in the LoRA-Pro paper). **Indeed, as shown in Figure 3 and Table 5,7, we observe that the preconditioning-based methods lead to worse loss convergence and adaptation performance compared to OD-LoRA.** We would like to highlight that in Theorem 1, optimal gradient approximation refers to the optimum when the subspaces of $A$ and $B$ are given, but it does not guarantee the global optimum. This implies that learning good subspaces of $A$ and $B$, which capture the principal components of the full gradient, is also essential. Thus, while projection-based methods elegantly achieve optimal gradient approximation with given subspaces, they may not effectively learn good subspaces, resulting in poorer gradient approximation with respect to the global optimum. By contrast, our new formulation, OD-LoRA, preserves the original gradients for $A$ and $B$, which may facilitate learning better subspaces. For an empirical validation, in Figure 4b, we compare OD-LoRA with these projection-based methods by measuring the alignment between the full fine-tuning gradient and the equivalent gradient. **The results show that while existing methods improve the alignment compared to the standard LoRA, the alignment is less than that of OD-LoRA.** We believe that investigating how to learn good subspaces to further improve gradient approximation quality is a valuable research direction.
>
> [1] Wang et al., “Lora-pro: Are low-rank adapters properly optimized?”, in ICLR2025.
>
> ## **2. LoRA adapters must be initialized to zero is not valid (W2)**
>
> As you mentioned, several methods initialize the LoRA parameters to non-zero by subtracting the offset from the pretrained weights, ensuring that the total weight update term is zero. The statement “$\Delta W_{OD-LoRA}$ must be zero to ensure that learning starts from the pretrained weight” also implies that the total weight update term, including offsets, must be zero at the beginning of training. If you mean the adapter itself (the part composed of LoRA parameters) rather than the total weight update term, it is possible to initialize the OD-LoRA adapter to non-zero using methods such as LoRA-GA or PiSSA. However, we believe that non-zero initialization of the LoRA adapter undermines the modularity of LoRA, since both the offset and LoRA parameters are required for the module to function. This approach incurs twice the memory cost for saving the trained LoRA modules and may create complications when merging multiple task-specific LoRA modules [2,3] with different offsets. **Therefore, we choose to initialize the adapter to zero to preserve the modularity of LoRA.**
>
> [2] Gu et al., "Mix-of-Show: Decentralized Low-Rank Adaptation for Multi-Concept Customization of Diffusion Models", in NeurIPS2023.
>
> [3] Zhao et al., "Merging LoRAs like Playing LEGO: Pushing the Modularity of LoRA to Extremes Through Rank-Wise Clustering", in ICLR2025.
>
> ## **3. Gradient misalignment during early training (W2)**
>
> In Algorithm 1, the standard LoRA formulation is applied only during the first $T$ iterations in the initialization phase (L274). After the initialization phase, we start from zero weight updates (i.e., $\Delta W_{OD-LoRA} = 0$) and apply the OD-LoRA formulation to achieve optimal gradient alignment (L284). **Consequently, our method ensures optimal gradient alignment throughout all training iterations.**

---

> ### Author Response · Authors · 2025-11-24
>
> ## **4. Comparison with related works only using LLaMA2-7B (W4)**
>
> In Table 5 and Table 7 in the revised version, **we include the results of gradient-based methods trained on both Gemma-2B and LLaMA-3-8B. As mentioned above, we also include the missing related works that you mentioned.** We reproduced them using the same experimental setup as detailed in Section 4.1 and Table 9. In Figure 3 and Figure 4b, we also compare OD-LoRA with existing methods in terms of gradient approximation quality and training loss convergence. **These results show that OD-LoRA achieves better alignment between the full gradient and the equivalent gradient and faster convergence toward lower loss, leading to better adaptation performance.** Please refer to Section D for further discussion.
>
> ## **5. Impact of optimizer momentum (Q1)**
>
> We understand your question in two possible ways.
> - First, if you are asking whether the proposed method updates the optimizer momentum according to the gradients for A and B, the answer is yes. As discussed earlier, our method achieves optimal gradient approximation without manual adjustment of gradients for A and B, ensuring that these gradients are automatically tracked by the optimizer. In contrast, methods that manually adjust the gradients for A and B require manually updating the optimizer momentum. This represents a meaningful distinction between our method and existing approaches.
> - Secondly, if you are asking whether the proposed method accounts for the impact of optimizer momentum on the full fine-tuning gradient, the answer is no. Our theoretical analysis assumes pure gradient descent without momentum. We agree that optimizer momentum can alter the effective full fine-tuning gradient, making our theoretical results suboptimal. One possible approach would be to track the momentum of the full gradient and use the modified gradient for updating the LoRA parameters, but this is computationally expensive. This limitation applies to both our work and related methods. We believe that extending our theories to momentum-based optimizers and developing a corresponding LoRA formulation would be a valuable direction for future research.
>
> **6. Typo (Q2)**
> Thank you for notifying us of the typo. We have corrected it in the revised version.

---

> > ### Comment · Reviewer_9ab4 · 2025-11-26
> >
> > I acknowledge that I have read the authors’ response, but I still retain my concerns.
> >
> > - **The theoretical justification for why OD-LoRA is fundamentally superior to gradient-approximation methods such as [1] and [2] remains insufficient.** This is particularly critical because OD-LoRA employs a similar preconditioning structure to these prior works, yet the authors do not clearly articulate where the theoretical advantage originates in Appendix D.
> > - **Regarding the initialization period, the authors note that OD-LoRA does not achieve alignment in the first T steps.** However, prior work such as LoRA-GA emphasizes that the initial steps are precisely where gradient misalignment is most harmful. The discrepancy between OD-LoRA’s behavior in early iterations and the claims of improved alignment therefore requires further justification.
> >
> > [1] Zhang F, Pilanci M. Riemannian preconditioned LORA for fine-tuning foundation models. ICML 2024.
> >
> > [2] Yu X, Wang Y, Chen J, et al. AltLoRA: Towards Better Gradient Approximation in Low-Rank Adaptation with Alternating Projections. Neurips 2025.

---

> ### Author Response · Authors · 2025-12-03
>
> We appreciate the reviewer’s response. Here are the responses to the unresolved concerns.
>
> ## **1. Theoretical justification of the difference between OD-LoRA and preconditioning-based methods [1,2]**
>
> As previously mentioned, although both OD-LoRA and existing methods achieve optimal gradient approximation by projecting the full fine-tuning gradient onto the low-rank subspace defined by the LoRA matrices, **they fundamentally differ in the update rules for the LoRA matrices ($A$ and $B$).**
> As shown in Equation 11 and Equation 12, existing methods manually **modify the original gradients** for $A$ and $B$ by multiplying the Gram matrices of them (i.e., preconditioner), whereas OD-LoRA's new formulation allows for the optimal gradient approximation **by construction without modifying the original gradients.**
>
> Since the quality of the approximation depends on how well the learned subspaces evolve to align with the full gradient over time, these distinct update rules result in divergent optimization trajectories and, consequently, different final model performance. **To further justify this distinction, in Section D.1, we add a theoretical analysis of how the different update rules for the LoRA parameters influence the training dynamics from the perspectives of subspace learning and convergence rate.**
>
> - In Section D.1.1, we evaluate the effectiveness of each method in learning desirable subspaces. We define the Signal-to-Noise Ratio (SNR) of the parameter update (Definition 3) and demonstrate that preconditioning-based methods inherently scale updates by the inverse of the singular values. In the context of LoRA, this operation disproportionately amplifies noise components associated with small singular values, thereby distorting the intrinsic SNR and hindering the subspace from evolving toward the dominant gradient direction. **In contrast, OD-LoRA preserves this intrinsic ratio because it structurally decouples the gradient magnitude from the singular values (Theorem 4), allowing the subspaces to align with the dominant gradient direction.**
>
> - In Section D.1.2, we compare the convergence rates of OD-LoRA and preconditioning-based methods. Our analysis shows that existing methods suffer from instability (requiring small learning rates) and slow convergence as the condition number (i.e., the ratio of the largest to the smallest singular value) of the LoRA matrices increases. **In contrast, OD-LoRA is independent of the condition number, enabling stable and fast convergence (Theorem 5).**
>
> **These theoretical analyses align with the experimental results in Section D.2**, which are presented in the previous revision.
> The results show that OD-LoRA achieves better alignment between the full fine-tuning gradients and the approximated gradients (Figure 4b) and converges faster to lower loss values (Figure 3).
>
> [1] Zhang F, Pilanci M. Riemannian preconditioned LORA for fine-tuning foundation models. ICML 2024.
>
> [2] Yu X, Wang Y, Chen J, et al. AltLoRA: Towards Better Gradient Approximation in Low-Rank Adaptation with Alternating Projections. Neurips 2025.
>
> ## **2. Suboptimal alignment during initialization phase**
>
> It is worth emphasizing again that after the initialization phase, we set the weight update term to zero and begin training from scratch using our formulation $\Delta W_{\text{OD-LoRA}} = s(Q_B A + B Q_A^\top)$ (L284), which guarantees optimal gradient alignment (Theorem 3). **Thus, our method ensures the optimal gradient alignment during all iterations.**
>
> We provide several important notes regarding the initialization phase to address potential confusion.
>  - The initialization phase is for identifying good initial subspaces ($Q_A$ and $Q_B$) for better gradient approximation during the main training phase.
> - There are $N$ cycles within the initialization phase where each cycle corrects $Q_A$ and $Q_B$ obtained from the previous cycle.
> - Only for the first cycle, we adopt the standard LoRA formulation to roughly initialize $Q_A$ and $Q_B$ (L274), and we adopt OD-LoRA formulation for the subsequent cycles (L276-L277).
>
> Thus, using the LoRA formulation for the first $T$ iterations in the initialization phase only affects the initialization of $Q_A$​ and $Q_B$ and does not imply that our method fails to achieve optimal gradient alignment during the main training phase.
> Although one might be concerned that updating $Q_A$​ and $Q_B$ with the LoRA formulation in the first cycle could be harmful, **this step serves only as a coarse initialization, and potential negative effects are diluted by the subsequent cycles that use the OD-LoRA formulation.**

---

### Official Review · Reviewer_94iS · 2025-10-30

**Soundness:** 3
**Presentation:** 3
**Contribution:** 3
**Rating:** 8
**Confidence:** 3

**Summary:**

This paper provides a theoretical analysis of LoRA, demonstrating that LoRA’s use of low-rank matrices A and B attempts to simultaneously achieve two conflicting objectives: (1) accurately approximating the full gradient, which requires uniform singular values, and (2) flexibly representing weight updates, which requires the singular values to be freely learned. To resolve this conflict, the paper proposes the OD-LoRA method, which decouples low-rank gradient approximation from the learning of singular values by utilizing an orthogonal basis defined by the LoRA subspace. Experimental results across multiple tasks show that OD-LoRA achieves superior performance compared to existing approaches.

**Strengths:**

1.The paper clearly and theoretically reveals the inherent trade-off in LoRA between accurate gradient approximation and weight representation capability.

2.It proposes the OD-LoRA method, which leverages an orthogonal basis defined by the LoRA subspace to decouple low-rank gradient approximation from the learning of singular values, effectively resolving the aforementioned dilemma.

3.Extensive experiments on multiple benchmark datasets demonstrate the superiority of the proposed method compared to LoRA and its variants.

**Weaknesses:**

1.In OD-LoRA, the initialization stage involves two hyperparameters, N and T, which are fixed in the experiments. It would be helpful to include ablation studies or sensitivity analyses to verify how different choices of these values affect performance.

2.Since LoRA-based fine-tuning often suffers from catastrophic forgetting, it would be valuable to investigate whether OD-LoRA also exhibits similar issues after fine-tuning.

**Questions:**

Please refer to the Weakness section.

---

> ### Author Response · Authors · 2025-11-24
>
> We appreciate your time and effort in reviewing our work. We are glad to receive such positive reviews regarding the clear theoretical structure, the proposed method, and its effectiveness. We also find the weaknesses you pointed out to be valuable, as they help us strengthen our work. We hope that our responses adequately address your concerns.
>
> ## **1. Sensitivity to N and T**
>
> In Section H and Table 7 of the revised version, we provide sensitivity analyses for N and T, the hyperparameters in the proposed initialization phase. The results show that as N or T increases, the performance of OD-LoRA tends to improve. However, the differences in performance are small, suggesting that the method is not highly sensitive to these hyperparameters.
>
> ## **2. Catastrophic Forgetting issue in OD-LoRA**
>
> We agree that the forgetting issue is important in the context of low-rank adaptation, as LoRA is often used in multi-task or continual learning scenarios. To assess OD-LoRA's robustness to forgetting, we merge two OD-LoRA weights trained on different tasks. Specifically, we fine-tune two Gemma-2B models using OD-LoRA on Code-Feedback and MetaMATHQA tasks. Let $\Delta W_C$ and $\Delta W_M$ denote the learned weight updates on the two tasks, respectively. Then, we obtain a merged weight update as $\Delta W_{merge} = \Delta W_C + \alpha \Delta W_M$ where $0<\alpha<1$. Using $\Delta W_{merge}$, we evaluate the performance on the Code-Feedback benchmark.
> The table below shows the results.
>
> | Method        | $\alpha=0$ |$\alpha=0.3$ |$\alpha=0.5$ |$\alpha=0.7$ |$\alpha=1$ |
> |---------------------|----------|----------|----------|----------|----------|
> | LoRA     | 31.10   |32.93 (+1.83)   |28.66 (-4.27)   |22.56 (-6.1)   |3.05 (-19.51)   |
> | OD-LoRA     | 34.15   |35.98 (+1.83)  |31.71 (-4.27)   |  28.05 (-3.66) | 9.76 (-18.29)|
>
> We observe that OD-LoRA shows slightly less performance degradation as $\alpha$ increases. These results may suggest OD-LoRA’s robustness to the forgetting issue, although the difference is not significant. While addressing forgetting is beyond the scope of this work, we believe that investigating the connection between improved gradient approximation and robustness to catastrophic forgetting would be a valuable research direction.

---

> > ### Comment · Reviewer_94iS · 2025-11-26
> > **Official Comment by Reviewer 94iS**
> >
> > Thank you for your detailed response. The additional experimental results demonstrate that the method is not highly sensitive to the hyperparameters N and T. Furthermore, the method outperforms LoRA in  mitigating catastrophic forgetting. The results have addressed my concern. Therefore, I maintain my positive rating for this paper.

---

### Official Review · Reviewer_LV8k · 2025-10-31

**Soundness:** 2
**Presentation:** 3
**Contribution:** 2
**Rating:** 2
**Confidence:** 4

**Summary:**

This paper proposes a new LoRA-based PEFT method called OD-LoRA. OD-LoRA addresses a fundamental dilemma between the gradient approximation and maintaining full representational capacity in low-rank weight updates. The authors first prove that LoRA achieves minimal gradient approximation error only when its low-rank matrices have uniform singular values, but such uniformity severely limits their representational capacity. To overcome this, OD-LoRA decouples the gradient approximation process from singular values by leveraging orthonormal bases of the LoRA subspaces to preserve the gradient direction.

**Strengths:**

* The idea of decoupling through orthonormal bases is conceptually elegant and technically novel.
* The paper is built on a strong theoretical foundation, which provides valuable insights into the geometric characteristics of LoRA.

**Weaknesses:**

* The frequency of computing the SVD of $\Delta W$ is not specified. Moreover, could you provide the time cost associated with the SVD steps in the ablation study? Intuitively, performing SVD frequently may yield more precise directions for $A$ and $B$, but it may also frequently reset the optimizer states, which may not help to improve the overall performance.

* In Algorithm 1, the initialization $A = 0, B = 0$ is confusing. Similarly, in Line 308, the statement “since $A = B = 0$” is unclear. If $A$ and $B$ are both set to zero in the algorithm, how are the backward and update steps for $A$ and $B$ performed? In the original LoRA design, the initializations for $A$ and $B$ are not both zero.

* Gradient-based methods (LoRA-GA, LoRA-Pro) are compared only on LLaMA2-7B with two tasks, which is not consistent with the main experimental results presented in Tables 1 and 2. I think comparing with these methods is important to show the effectiveness of the proposed method.

* Moreover, what are the learning rate and experimental settings for Table 5? If the settings used in Table 5 correspond to those shown in Table 9, then the comparison in Table 5 may be unfair. As shown in Table 9, OD-LoRA adopts a learning rate of 2e-4, 1 training epoch, and a batch size of 32. However, this configuration is not consistent with the original LoRA-GA paper, while the LoRA-GA row in Table 5 is exactly the same as that in Table 2 of the LoRA-GA paper.

**Questions:**

See questions in the weaknesses part.

---

> ### Author Response · Authors · 2025-11-24
>
> We appreciate your time and effort in reviewing our work. We are encouraged by your positive comments on the novelty of our method and its strong theoretical foundation. Also, we find that certain parts of our paper may have been unclear and could have led to misunderstandings. We have done our best to address the concerns raised and hope that our responses clarify any ambiguities and resolve your questions.
>
> ## **1. Clarification on the initialization strategy (W1, W2)**
>
> As presented in Equation 4, OD-LoRA formulates weight updates as $\Delta W_{OD-LoRA} = s(Q_B A + B Q_A^\top)$ where $Q_A$ and $Q_B$ denote the orthonormal bases of the row space of $A$ and column space of $B$. At the beginning of training, we observe that those subspaces are not yet well-defined and change rapidly due to the small norm of $A$’s rows and $B$’s columns (L300-L303). Thus, before the main training phase, we initialize $Q_A$ and $Q_B$ by running $N \times T$ training iterations.
> - (**SVD computation and its overhead.**) For all experiments, we set $N = 5$ and $T=10$ and SVD is performed at every $T$ iteration, resulting in five SVD computations (L279). During the main training phase, we compute the SVD of $\Delta W_{OD-LoRA}$ one more time at the $T$-th iteration to ensure $Q_A$ and $Q_B$ form the bases of the subspaces of $A$ and $B$ (L287). **Therefore, only $N+1=6$ computations of SVD are performed in our training algorithm.** (Please refer to Section H for the sensitivity analyses for $N$ and $T$.)  As detailed in Table 6 and Section 4.7, we observe that the initialization phase, which consists of 50 training iterations and five SVD computations, incurs about 1.5 % time overhead (0.03 hours out of 1.7 hours), which can be considered minor. Additionally, as you mentioned, we reset the optimizer states after updating $A$ and $B$ using the SVD results to remove any adverse impact on the optimizer states (L281 and L290). In the revised version (L318-L323), we clarify the time cost associated with the initialization phase.
> - (**$A=B=0$ is confusing.**) After initialization of $Q_A$ and $Q_B$, we set $A=0$ and $B=0$ to ensure $\Delta W_{OD-LoRA} = s(Q_B A + B Q_A^\top) = 0$ at the beginning of the training. As presented in L253-L254, the gradients for A and B are expressed as $sQ_B^TG$ and $sGQ_A$, respectively. **Thus, A and B can be updated even if $A=B=0$.** Note that $Q_A$ and $Q_B$ do not correspond to the orthonormal bases of the $A$'s row space and $B$'s column space during the first $T$ iterations in the main training phase. Please refer to L309-L316 for a detailed explanation.
>
> ## **2. Comparison with gradient-based methods. (W3, W4)**
>
> In Table 5 and Table 7 of the revised version, **we include the results of gradient-based methods trained on both Gemma-2B and LLaMA-3-8B. We also include several missing related works [1,2].** We reproduced them using the same experimental setup as detailed in Section 4.1 and Table 9. In Figure 3 and Figure 4b, we also compare OD-LoRA with existing methods in terms of gradient approximation quality and training loss convergence. These results show that OD-LoRA achieves better alignment between the full gradient and the equivalent gradient and faster convergence toward lower loss, leading to better adaptation performance. Please refer to Section D for further discussion.
>
> Additionally, to account for the fact that different methods may require different learning rates, **we also compare our method with them under various learning rates.** The table below shows the results using LLaMA-3-8B evaluated on GSM8K with rank 8.
>
> ||$\eta=1e-4$|$\eta=5e-5$|$\eta=2.5e-5$|
> |-|:-:|:-:|:-:|
> |LoRA-GA|75.21|73.84|73.54|
> |LoRA-Pro|72.93|71.49|66.41|
> |rsLoRA + ScaledAdamW|73.39|71.19|67.17|
> |AltLoRA|73.69|70.66|67.25|
> |OD-LoRA|76.62|75.74|74.45|
>
> The results clearly show that our method consistently outperforms existing methods across all learning rate settings.
>
> [1] Zhang et al., "Riemannian preconditioned lora for fine-tuning foundation models.", In ICML2024.
>
> [2] Yu er al., "Altlora: Towards better gradient approximation in low-rank adaptation with alternating projections.", in NeurIPS 2025.

---

### Official Review · Reviewer_iDJs · 2025-10-31

**Soundness:** 2
**Presentation:** 3
**Contribution:** 2
**Rating:** 4
**Confidence:** 3

**Summary:**

The paper argues that LoRA faces a dilemma between two roles: weight representation, as it must flexibly represent diverse low-rank updates, and gradient approximation, as its gradient corresponds to a low-rank projection of the full fine-tuning gradient. The authors claim this dilemma arises because the gradient is best approximated when the singular values of the LoRA factors are uniform, but this condition simultaneously limits representational capacity. To address this, they propose OD-LoRA, which decouples gradient computation from the singular values by using orthonormal bases obtained via QR decomposition. This reparameterization aims to achieve accurate gradient approximation and unrestricted weight representation, leading to slightly faster convergence and modest but consistent improvements over standard LoRA and recent variants across NLP and vision benchmarks.

**Strengths:**

1. Clear theoretical framing: The paper formalizes an internal tension in LoRA between representing flexible low-rank updates and approximating the full fine-tuning gradient, presenting the argument in a clean, mathematically consistent way.
2. Simple and efficient modification: OD-LoRA is easy to implement and incurs negligible computational overhead.
3. Comprehensive experiments: Evaluation spans language, reasoning, code, and vision tasks with several model sizes, showing the method's generality.
4. Consistent empirical gains: OD-LoRA typically converges faster and outperforms LoRA and recent variants (rsLoRA, DoRA, PiSSA) by small but steady margins.

**Weaknesses:**

1. Conceptual overreach of the “dilemma”: The paper’s central claim that LoRA’s non-uniform singular values cause a fundamental trade-off between gradient approximation and representational capacity is formally true but empirically unsubstantiated. There is no evidence that this dilemma actually affects LoRA’s training dynamics or performance in practice. The analysis basically assumes that better approximation of the full fine-tuning gradient improves performance, but this is not necessarily true. But why is full fine-tuning the gold standard when it often overfits or destroys pretrained representations? LoRA’s success arguably stems from restricting learning to low-dimensional subspaces rather than reproducing full fine-tuning behavior. Can you provide some empirical evidence that connects gradient misalignment to LoRA performance?

2. Questionable metric for gradient approximation: The cosine similarity between the full gradient and its low-rank projection is a weak and potentially misleading metric. It operates in euclidean space and measures direction only, not loss changes. This might be ignoring the fact that some directions are more important than others. I would suggest exploring alternative notions of "approximation" and justifying why the one chosen by the authors is more adequate here.

3. Limitations of the experimental evidence: while the experiments are extensive and carefully executed, they do not convincingly validate the paper’s central claims. The reported accuracy gains are modest and consistent but never traced to the proposed mechanism: the experiments measure only downstream performance, not whether LoRA actually suffers from the claimed trade-off or whether OD-LoRA resolves it. The ablation studies confirm internal consistency but remain correlational, as multiple confounding factors such as orthogonalization, initialization resets, and altered optimization dynamics could explain the improvements. Appendix D’s singular-value analysis merely shows that full fine-tuning produces anisotropic spectra, without comparing LoRA or OD-LoRA or linking spectral shape to performance. As a result, the empirical section supports that OD-LoRA works slightly better in practice but fails to establish why it does or whether the theoretical “dilemma” is a real limitation in existing LoRA methods.

**Questions:**

1. Can the authors provide empirical evidence that the proposed “dilemma” between representational capacity and gradient approximation actually arises during standard LoRA training?

2. Why should approximating the full fine-tuning gradient be considered desirable, given that LoRA’s efficiency often comes from deviating from full fine-tuning?

3. Are there theoretical conditions under which better gradient alignment provably improves convergence or generalization?

4. Could the authors show that OD-LoRA’s improved performance correlates with reduced gradient misalignment, rather than to unrelated optimization effects?

---

> ### Author Response · Authors · 2025-11-24
>
> We appreciate your time and effort in reviewing our work. We are motivated by your positive comments regarding our theoretical approach, simple yet effective method, comprehensive experiments, and consistent performance gains. Also, we find your constructive feedback very helpful for improving our work. We have done our best to address your concerns and hope that our responses adequately resolve them.
>
> ## **1. The necessity of improving gradient approximation quality in LoRA (W1, Q2, Q3, Q4)**
>
> We agree that full fine-tuning may lead to overfitting or forgetting, whereas LoRA can mitigate these issues by restricting learning to low-rank subspaces. We would first like to emphasize that our method still preserves the low-rank learning constraint of LoRA while improving the gradient approximation quality. From this low-rank-constrained learning perspective of LoRA, if the low-rank approximation of the full gradient is inaccurate, useful information from the full gradient may be lost, resulting in ineffective learning. To provide theoretical evidence, in the early part of Section 3.2 of the revised version, we show that **when the approximated gradient substantially deviates from the full fine-tuning gradient (in terms of inner product), the loss-decrease guarantee deteriorates, potentially leading to suboptimal learning.** To further support this argument, we also provide empirical evidence in Figure 4a, showing that the performance of LoRA and OD-LoRA is closely tied to the alignment between the full gradient and its low-rank approximation. The results show that as the rank of the LoRA matrices increases, the approximated gradient becomes more aligned with the full gradient, and OD-LoRA achieves significantly better alignment than LoRA. **These findings provide strong justification for the importance of improving gradient approximation quality in LoRA and indicate that the performance gain of OD-LoRA is correlated with its better gradient approximation.** Please refer to Section E for further discussion.
>
>
> ## **2. Metric for gradient approximation (W2)**
>
> We agree that the original metric, cosine distance, lacks a clear justification as a measure of gradient approximation error. In the revised version, we evaluate gradient approximation quality from the perspective of the loss-decrease guarantee. Specifically, Equation 3 suggests that a larger inner product between the full gradient and its low-rank approximation ensures a steeper loss decrease. While the original metric partially reflects inner product, **we adopt the inner product directly as a more general definition of gradient approximation quality.** Note that the proof in Section A shows that this modification does not affect our theoretical results, since we assume the energy of LoRA parameters A and B is given at a certain training iteration. We believe this revision makes our theoretical claims clearer and more convincing. We appreciate your suggestion.
>
> ## **3. Empirical evidence on the dilemma in LoRA (W1, W3, Q1)**
>
> In Theorem 2, the dilemma in LoRA implies that the standard formulation $\Delta W_{\text{LoRA}} = sBA$ cannot simultaneously achieve the full representational capacity of weight updates and the condition for optimal gradient approximation quality in Theorem 1. For simplicity, let $R_{\text{cap}}$ and $R_{\text{approx}}$ denote these two requirements, respectively.
> We first note that standard LoRA satisfies $R_{\text{cap}}$ but not $R_{\text{approx}}$. As discussed above, $R_{\text{approx}}$ is essential for improving the training dynamics of LoRA.
> To examine whether the dilemma in LoRA has negative impacts, we demonstrate that simultaneously satisfying both requirements is crucial for enhancing the training dynamics of LoRA and its adaptation performance.
> In Table 4 of the revised version, we introduce 'rsLoRA w/ Optimal $\tilde{G}$', a variant of rsLoRA (LoRA with the same scaling factor $s$ as OD-LoRA) that satisfies $R_{\text{approx}}$ while sacrificing $R_{\text{cap}}$. To isolate its effect, we remove the proposed initialization phase for this variant. Although this variant improves gradient alignment (right side of Figure 2), it does not improve performance (Table 4) and loss convergence (left side of Figure 2). Furthermore, the results of 'OD-LoRA w/ $Cap_{X}(\Delta W)< \infty$' and 'OD-LoRA w/ Suboptimal $\tilde{G}$' show that intentionally breaking either $R_{\text{cap}}$ or $R_{\text{approx}}$ in OD-LoRA leads to deteriorated loss convergence and worse performance. **These results demonstrate the negative impact of the dilemma in LoRA: while both $R_{\text{cap}}$ and $R_{\text{approx}}$ are essential, the standard LoRA cannot achieve them simultaneously.** In contrast, OD-LoRA’s formulation achieves both requirements, enhancing the training dynamics of LoRA (Figure 2) and, consequently, the adaptation performance. Please refer to Section 4.5 and Section I for details.

---

> ### Author Response · Authors · 2025-11-24
>
> ## **4. Performance gain (W3)**
>
> We would like to emphasize that **the performance gain should be evaluated relative to LoRA and full fine-tuning, since different tasks and models have varying adaptation difficulty.** Comparing results on Gemma-2B and LLaMA3-8B, the performance gap between LoRA and full fine-tuning is much smaller for LLaMA3-8B, which benefits from stronger pre-trained knowledge and is easier to adapt to downstream tasks. Consequently, the performance improvement of OD-LoRA is relatively modest in LLaMA3-8B experiments compared to Gemma-2B. In contrast, results on Gemma-2B show that OD-LoRA outperforms the other methods by a large margin, significantly reducing the performance gap between LoRA and full fine-tuning. For instance, on the HellaSwag benchmark (Table 1), while LoRA and its variants lag substantially behind full fine-tuning, OD-LoRA achieves performance comparable to full fine-tuning. Thus, we would like to highlight the importance of evaluating the effectiveness of our method in this context.
>
>
> **Note: Confusion regarding ‘singular values’**
>
> We suspect that some misunderstanding regarding singular values may have arisen from a confusing part of the paper. In particular, in Table 4, the second column ‘Uniform $\sigma$’ represents the case where the equivalent gradient $\tilde{G} = s^2 (GA^\top A + BB^\top G)$ satisfies the optimal condition in Theorem 1, uniform singular values of $A$ and $B$. While this condition influences the singular values of the weight updates in standard LoRA, it does not affect the singular values of the weight updates in OD-LoRA. This is the core idea of OD-LoRA: decoupling the singular values of the weight updates from those of the equivalent gradient (see L239–L247 for further discussion). Consequently, in OD-LoRA, the relationship between the singular values of the weight updates and the resulting performance gain is not meaningful. We present the singular value distribution from full fine-tuning to show that a desirable weight update matrix typically exhibits non-uniform singular values. In turn, standard LoRA tends to learn weight updates with a similar singular value distribution, as shown in Figure 6, leading to inaccurate gradient approximation as demonstrated in the first bullet point of Theorem 2. To eliminate the confusion, we have revised Table 4 and the corresponding main text. If you already understood this point correctly, please feel free to ignore this explanation.

---

### Author Response · Authors · 2025-12-03
**Summary of reviews and our responses**

We appreciate the Area Chair’s effort in handling the unexpected challenges in the review process.
To support an efficient evaluation, we summarize below both the strengths highlighted by the reviewers and the remaining concerns with our responses.

## **1. Strengths**

| Strength | Mentioned by |
|----------|--------------|
| Clear and strong theoretical foundation | iDJs, 94iS, 9ab4, LV8k |
| Conceptual novelty | LV8k, 94iS, 9ab4 |
| Valuable insights | LV8k, 9ab4 |
| Simple and efficient method design | iDJs |
| Comprehensive and diverse experiments | iDJs, 94iS, 9ab4 |
| Consistent performance gains | iDJs, 9ab4 |

## **2. Concerns & Responses (sorted by importance)**

*Note: Concerns that have already been addressed or clarified are excluded.*

### **Concerns**

| # | Concern | Reviewer |
|---|---------|---------|
| 1 | Comparison with related works | LV8k, 9ab4 |
| 2 | Empirical evidence on the negative impact of dilemma in LoRA| iDJs |
| 3 | Necessity of improving gradient approximation quality in LoRA | iDJs |
| 4 | Justification of proposed measure for gradient approximation quality | iDJs |
| 5 | Confusion on the initialization method regarding overhead and gradient computation | LV8k |
| 6 | Gradient misalignment during the initial phase | 9ab4 |
| 7 | Performance gain | iDJs |


### **Our Responses**

**1. Comparison with related works**

- (**Revision**) Beyond the comparison using LLaMA2-7B, we added comparisons on Gemma-2B and LLaMA3-8B (Table 5,7), including loss convergence (Figure 3) and gradient approximation quality (Figure 4b). (This concern is resolved for Reviewer 9ab4)

- (**Revision**) Regarding the comparison with preconditioning-based methods (raised by 9ab4), we highlighted that to achieve the same goal (optimal gradient approximation), existing methods manually modify the gradients for LoRA matrices, whereas OD-LoRA achieves it by construction without modifying the original gradients. We have theoretically shown that this difference allows OD-LoRA to achieve more effective subspace learning (Section D.1.1) and better convergence rate (Section D.1.2).

*Note: Reviewer 9ab4 mentioned that our previous responses regarding the comparison with preconditioning-based methods were insufficient. This is the subsequent response and revision.*

**2. Empirical evidence on the negative impact of dilemma in LoRA**

We clarified that the dilemma in LoRA indicates that LoRA cannot achieve the two requirements (full representational capacity of weight updates and optimal gradient approximation) at the same time, which is theoretically proven. We highlighted that the ablation studies demonstrate the negative impact of the dilemma by showing that loss convergence and final performance improve only when the two requirements are satisfied simultaneously. (**Revision**) For fairer ablation studies, we added one additional variant.

**3. Necessity of improving gradient approximation quality in LoRA**

(**Revision**) In L164-L179, we have shown that as the approximated gradient deviates from the full fine-tuning gradient, the loss-decrease guarantee deteriorates. For further justification, we conducted experiments (Figure 4a) showing that a positive correlation between gradient approximation quality and the resulting adaptation performance.

**4. Justification of proposed measure for gradient approximation quality**

(**Revision**) In L164-L179, we propose evaluating the quality of gradient approximation from the perspective of loss-decrease guarantee. Based on this, we extend the original metric (cosine similarity) to the inner product for a more general definition of gradient approximation quality. (Note: this does not affect our theoretical results and proposed method.)

**5. Confusion on the initialization method regarding overhead and gradient computation**

We clarified that the proposed initialization method incurs negligible overhead ((**revision**) L318-L323 and Table 6) and that the gradients can be computed after the initialization phase (L253).

**6. Gradient misalignment during the initial phase**

We clarified that our method achieves the optimal gradient alignment during the main training phase.

**7. Performance gain**

We emphasized that OD-LoRA’s performance gains depend on task/model difficulty: modest for easier-to-adapt models like LLaMA3-8B, but substantial for harder models like Gemma-2B, significantly closing the gap with full fine-tuning (Tables 1,2,3).

---

### Meta-Review · Area_Chair_fVbU · 2026-01-04

**Summary:**

The reviewers had the following concerns:
1. Insufficient theoretical and empirical justifications of the proposed method. The reviewers were concerned about how the identified dilemma in LoRA optimization directly affects the model performance, and how the proposed method can successfully resolve the dilemma.
2. Missing comparison with other existing projection-based methods in terms of the technical novelty.
3. Missing discussions/analyses on the effect of gradient misalignment during the initialization phase (standard LoRA training) in the proposed method.
4. There are several technical questions regarding the initialization of A, B matrices, the time cost of computing SVDs, etc.
5. Insufficient empirical comparisons with the gradient-based baselines (LoRA-GA, LoRA-Pro).

**Reviewer Concerns:**

Regarding the first concern, the author provided additional theoretical justification of the necessity of a good gradient approximation from a loss-approximation perspective, and further clarified how the proposed method addresses the dilemma. The responses seem reasonable to me, thus I view this concern somewhat addressed.

Regarding the second concern, the authors provided additional theoretical analyses to articulate the advantage of the proposed method against existing ones. I have briefly read those sections and personally found them unconvincing: The authors proposed some notion of signal-to-noise ratio to justify the advantage of their proposed update. However, at a high-level, viewing some part of the gradient as "noise" is fundamentally against the idea of preferring a full approximation of the gradient. Therefore, the argument seems to be invalid without further refinement. I view the second concern still outstanding.

Regarding the third concern, the authors have noted in the rebuttal that "The initialization phase is for identifying good initial subspaces for better gradient approximation", but the reviewers' concerns are precisely that the "good initial subspaces" might not be obtained if using standard LoRA in the initialization phase, which the authors did not respond to in the rebuttal; and I agree with this potential issue and view the third concern still outstanding.

The forth and fifth concerns are addressed in the rebuttal

**Reviewer Scores:**

I think reviewers 94iS and 9ab4 would keep their respective scores (8 and 4); in particular, the primary concerns from the reviewer 9ab4 were not addressed. It is possible that the reviewer iDJs increases their score, depending on whether they think the first concern is addressed. I believe the reviewer LV8k is likely to increase their score (2 to 4) given that their concerns were mostly addressed (it is hard to predict the increased score since it also depends on how reviewer LV8k would agree with other reviews).

Overall, the paper receives borderline scores. I view the outstanding concerns (technical novelty, issues with the initialization phase) as major weaknesses in this paper, thus I recommend rejection.

---

### Decision · Program_Chairs · 2026-01-26

Reject